# Antifungal alternation can be beneficial for durability but at the cost of generalist resistance

Agathe Ballu [1], Philomène Despréaux[1], Clémentine Duplaix[1], Anne Dérédec[1], Florence Carpentier [2,3,4] & Anne-Sophie Walker [1,4✉]

The evolution of resistance to pesticides is a major burden in agriculture. Resistance management involves maximizing selection pressure heterogeneity, particularly by combining active ingredients with different modes of action. We tested the hypothesis that alternation may delay the build-up of resistance not only by spreading selection pressure over longer periods, but also by decreasing the rate of evolution of resistance to alternated fungicides, by applying an experimental evolution approach to the economically important crop pathogen *Zymoseptoria tritici*. Our results show that alternation is either neutral or slows the overall resistance evolution rate, relative to continuous fungicide use, but results in higher levels of generalism in evolved lines. We demonstrate that the nature of the fungicides, and therefore their relative intrinsic risk of resistance may underly this trade-off, more so than the number of fungicides and the rhythm of alternation. This trade-off is also dynamic over the course of resistance evolution. These findings open up new possibilities for tailoring resistance management effectively while optimizing interplay between alternation components.

[1] Université Paris-Saclay, INRAE, UR BIOGER, 91120 Palaiseau, France. [2] Université Paris-Saclay, INRAE, UR MaIAGE, 78350 Jouy-en-Josas, France. [3] AgroParisTech, 91120 Palaiseau, France. [4] These authors contributed equally: Florence Carpentier, Anne-Sophie Walker. ✉email: anne-sophie.walker@inrae.fr

Humanity faces a number of crucial challenges, including obtaining sufficient feed, food, fuel, and fiber. The protection of crops still relies heavily on the use of pesticides to control diverse pests. However, the efficacy of these active ingredients (AIs) has been compromised by the generalization of their use, leading to the rapid and widespread evolution of resistance[1,2]. Resistance can be seen as phenotype optimization in response to the selection imposed by pesticides and has provided many examples of contemporary adaptive evolution[3]. The costs engendered by the evolution of resistance in pathogens, arthropods, and weeds amount to billions of US dollars each year[4,5] and entail hidden costs, due to greater pesticide use[6] and impacts on biodiversity[7,8]. The use of smart strategies for preventing and delaying the evolution of resistance is, thus, crucial for sustainable agriculture.

Diverse AIs, with different modes of action, are often available for controlling pests and pathogens. Strategies can therefore be developed in which these AIs are skilfully deployed over space and time so as to limit the evolution of resistance[9,10]. More precisely, the optimization of sequences, mixtures, alternation, mosaic, or dose strategies for pesticides involves maximizing the heterogeneity of selection pressures exerted on the pathogen and pest populations[11–15]. The alternation (also known as cycling, rotation, or periodic application) of unrelated AIs was first proposed by Coyne in 1951[16] and provides multiple means of intergenerational killing, maximizing the probability of killing the offspring of resistant individuals[10]. The efficacy of alternation for preventing resistance remains a matter of debate, as ranking discrepancies between modeling and empirical data have been noted, for control strategies for insects, weeds, and pathogens[10,17–19]. The biological traits of the organisms concerned (e.g. the reproduction mode, the ploidy of the species, the dominance of the resistance alleles, the duration and overlapping of generations with regard to pesticide exposure, etc.) may partly account for this divergence, as these traits were not systematically made explicit in the mathematical models, and different numbers of generations were considered in the two approaches. However, the variability of the performance of alternation, and, indeed, of other strategies, raises questions about which components of the strategy are the most relevant for delaying the evolution of resistance. Indeed, the heterogeneity of selection in an alternation strategy depends on how many AIs are alternated, the nature of the AIs concerned, and the pattern of alternation adopted.

Another limitation is that strategies are mostly evaluated on the basis of their quantitative impact on resistance evolution (e.g. resistance frequency, the proportion of healthy crops, time to reach a certain amount of resistance, the effective life of the pesticide), and more rarely on the basis of the characteristics of the resistant individuals (e.g. resistance intensity; the pattern of cross-resistance; nature and a number of resistance mechanisms). This may be because the genetic basis of resistance is rarely known, in empirical studies, or made explicit, or because mathematical models assume the resistance trait to be specialist[20,21]. Nevertheless, the diversity of resistance mechanisms observed in microbes, animals, and plants is considerable. The evolutionary origins of these mechanisms may also differ[22]. Target site resistance (TSR) affects the structure and/or expression of the target gene of the pesticide by one or several changes in its coding sequence and/or promoter or regulatory gene, resulting in cross-resistance to AIs with the same mode of action (MoA). This type of resistance is considered to be a specialist adaptation, even if levels of cross-resistance for TSR may vary between mutations and MoAs. By contrast, non-target site resistance (NTSR) involves the regulated expression of one or multiple genes involved in AI transport, bypass, detoxification, efflux, or sequestration. This generally leads to cross-resistance to AIs with

different MoAs and is usually considered a generalist trait[23–26]. Theory, with support from experimental evolution experiments, suggests that specialists evolve in homogeneous environments, whereas generalists are more likely to evolve in heterogeneous habitats[27]. Thus, in terms of adaptation to pesticides, we would expect generalist resistance to be more readily selected with increasing heterogeneity of selection pressure, as in alternation, mixture, or mosaic strategies, in contrast to the continuous use of a single AI. This hypothesis has been validated for herbicide mixtures, for which increasing use of different MoAs was found to be associated with lower levels of TSR and higher levels of NTSR (i.e. detoxification) in the economically important weed blackgrass, in a national survey in the UK[28]. A similar result was found in the clinical environment, where generalist resistance has also been shown to be more frequently selected in situations in which antibiotics are alternated[29]. Overall, these studies highlight the importance of treatment frequency relative to the pathogen life cycle[30,31]. The question of the efficacy of alternation strategies thus remains unresolved for pesticides, as treatment frequency may result in the interval between treatments being much longer than the generation times of pests, particularly for some pathogens.

Here, we use the ascomycete fungus *Zymoseptoria tritici*—the cause of septoria leaf blotch (STB), the most devastating foliar disease of wheat in Europe[32,33]—as a general model to determine how the components of fungicide alternation drive the quantitative and qualitative performance of this strategy. This highly adaptive pathogen, with large and diverse populations and dual reproduction mode, has evolved resistance in the field to four different unisite modes of action in Western Europe, resulting in contrasting decreases in field efficacy depending on AIs and their combination[34–40]. Resistance dynamics also differ according to MoAs, as demonstrated quantitatively for France[41,42]. *Z. tritici* exhibits TSR, determined by various specific target-site mutations, that can be associated with target overexpression[40,43–45] but also with NTSR via enhanced efflux. This latter generalist mechanism causes weak multidrug resistance (MDR) when not associated with TSR and affects at least three modes of action[43,46]. These biological features and field background make *Z. tritici* a textbook model for dissecting the adaptation of populations under contrasting selection pressures and reasoning about general principles. We made use of the haploid yeast-like easily culturable form of this pathogen (i.e. blastospores)[47] in a liquid medium, in an experimental evolution approach[48] based on the selection of resistance under regimes mimicking alternation. The selection patterns were designed such that three components of alternation —the intrinsic risk of resistance to AIs, the number of AIs alternated, and the alternation rhythm—could be disentangled, to assess their respective impacts on resistance dynamics and on the resistance mechanisms selected. We found that the combination and interplay between alternation drivers clearly guided the evolutionary trade-off between the quantitative (i.e. the overall and fungicide-specific resistance evolution rates) and qualitative (i.e. the degree of generalism of evolved strains) performances of alternation, opening up new perspectives for the informed tailoring of this strategy in the field, for the first time in a phytopathogenic fungus.

## Results

**Fungicide alternation has a neutral-to-beneficial effect, slowing the overall evolution rate of resistance.** We observed the dynamics of *Z. tritici* resistance to benzovindiflupyr (B), carbendazim (C), and prothioconazole-desthio (P), here representing three modes of action (SDHIs, benzimidazoles, and DMIs, respectively), after experimental evolution in 56

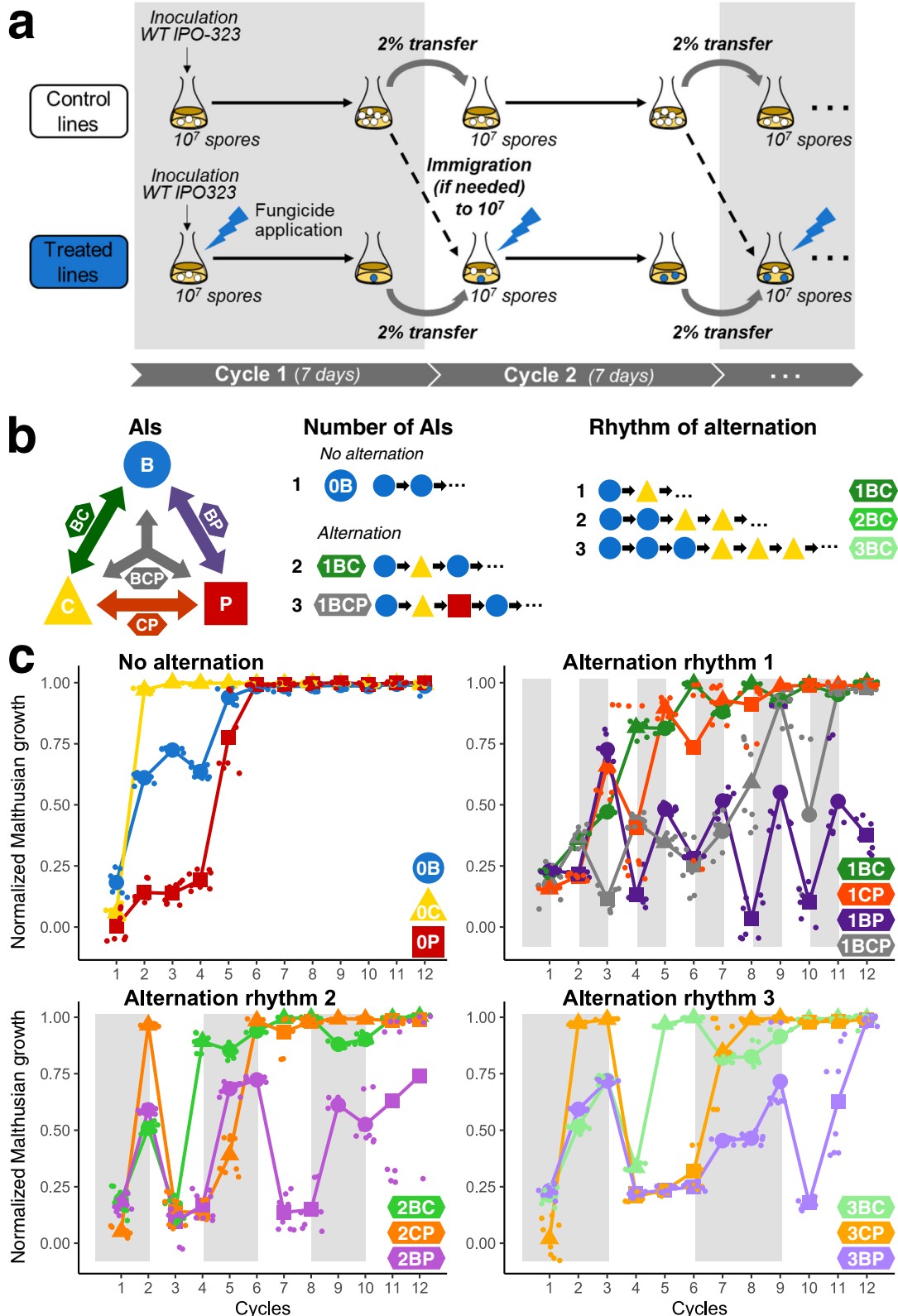

independent lines subjected to 14 regimes of continuous or alternating B, C and P, including B, C or P at their $EC_{95}$ selection doses (Fig. 1a and b). Mean normalized Malthusian growth $M_{it}^n$ (referred to hereafter as the overall resistance evolution rate $\rho$) was used as a proxy for increasing resistance. Resistance establishment was declared when $\rho$ reached at least 90% of that of the control line. Resistance evolved in all

populations continuously exposed to B, C, or P, and was established after three, six, and six cycles, respectively. Resistance also evolved in lines treated with these three fungicides in alternation, regardless of the alternation rhythm (i.e. the number of successive cycles undergoing the use of the same AI) and the number of fungicides alternated, although resistance never established, even after 12 cycles, in some situations

**Fig. 1 Experimental evolution of fungicide resistance under treatments with contrasting temporal heterogeneities. a** The sensitive strain IPO-323 of *Z. tritici* was used to find 56 lines, with each Erlenmeyer flask inoculated with $10^7$ spores. Fungicides were added to the treated lines to mimic 14 patterns of selection. After 7 days, 2% of the population was used to inoculate fresh medium to start the next cycle, supplemented, if necessary, by immigration from the untreated control line to achieve a total of $10^7$ spores. This procedure was repeated for 12 cycles in four replicate lines per fungicide treatment. **b** Three fungicides and all the possible combinations (as described by the arrows and their specific labels and colors) of two and three fungicides were studied: carbendazim (C; yellow; high risk), benzovindiflupyr (B; blue; moderate risk), prothioconazole-desthio (P; red; low risk). For each pair of fungicides, three alternation rhythms were tested (1, 2, or 3, depending on the number of cycles with the same selection). **c** Some fungicide alternations effectively attenuated resistance dynamics. For each treated line, normalized Malthusian growth, estimated at the end of the cycle, is shown as a function of time. Colors indicate the treated lines and symbols (triangle, circle, and square) indicate the fungicide applied (C, B, and P, respectively) during a given cycle. For each cycle and selection regime, these large symbols represent the mean of observations, with individual observations shown as light dots. Overall, resistance evolved and became established within six cycles in lines with continuous exposure, whereas establishment in lines subjected to alternating exposure was either delayed or had not yet been reached after 12 cycles (1BP, 2BP).

including medium- to low-risk AIs (Fig. 1c). For lines undergoing direct selection (with a single compound), $\rho$ was 1.1 times higher with C than with B and 1.4 times higher with C than with P; these significant differences reflected the hierarchy of the relative intrinsic risks of resistance associated with benzimidazoles (high; C), SDHIs (medium; B) and DMIs (low; P) (Supplementary Fig S1). The overall resistance evolution rates of the lines were modified by alternation regimes, being significantly lower, by a factor of 1.1–1.3, for alternation regimes containing carbendazim than for continuous exposure to this high-risk fungicide. Alternation thus reduced the risk of resistance to C to a level similar to that observed for medium- and low-risk fungicides in conditions of continuous exposure. The benefits of alternation, by comparison to the use of single AIs, were even more pronounced, with a 25–58% decrease in resistance selection (as estimated by $\rho$) when low- and medium-risk fungicides were alternated or when fungicides of the three categories were alternated. Thus, depending on the fungicides alternated, alternation had either a neutral effect or decreased the overall resistance evolution rate relative to single fungicides in continuous use.

**The risk of resistance inherent to the fungicide is the key driver for tailoring alternation**. Alternation may reduce the selection of resistance to a particular fungicide by limiting the total time of exposure to this fungicide, but also by modifying the overall resistance evolution rate $\rho$, potentially through the diversification of selection pressure. We investigated the potential effect of multi-directional selection in alternation, separating this effect from that of the decrease in exposure time, by comparing $\rho$ only for time segments corresponding to fungicide f exposure ($\rho_f$, referred to hereafter as the fungicide-specific resistance evolution rate), and by alternation component (Fig. 2a). The impact of alternation on $\rho_f$ was significantly dependent on the intrinsic risk of resistance associated with the alternating partner. Indeed, the selection of resistance to the high-risk fungicide was decreased (by 9%) only if the fungicide concerned was alternated with the low-risk fungicide. Only alternation with the low-risk fungicide was able to decrease (by 22%) the selection pressure associated with the use of the medium-risk fungicide. Medium- and high-risk alternation partners had neutral or detrimental (+82%) impacts, respectively, on the rate of selection of resistance to the low-risk fungicide. Overall, the selection capacity of a fungicide was decreased only by alternation with a fungicide of sufficiently lower risk. Thus, decreases in selection pressure in alternation were always achieved at the expense of the selection capacity of the partner with the lower intrinsic risk of resistance. This conclusion also extended to strategies involving an alternation of three fungicides. The duration of continuous exposure to the same fungicide (alternation rhythm over the cycles) had a lesser effect, increasing the resistance evolution rate only for the longest

sequences (three cycles) and for only two of the 18 possible comparisons. Finally, the fungicide-specific resistance evolution rate was driven principally by the inherent resistance risk of its alternation partner relative to its own resistance risk, and secondarily, by other alternation components, such as the number of fungicides alternated and the alternation rhythm.

**Resistance selection differs over the course of evolution, reflecting contrasting shifts in the interplay between fungicides and phenotype-adaptive landscapes**. Changes in the frequency of resistance in a population typically follow the sigmoid curve, with an emergence phase of low resistance (<15%; here, the lowest frequency reliably detectable by measuring the change in $OD_{405}$ in our experimental conditions), a selection phase (here, resistance rates between 15% and 90%, chosen as a convention) and an established-resistance phase (here, over 90%)[13]. Cox survival analysis was used to investigate the effect of alternation components on the exposure time (number of cycles with a given fungicide) required to reach a given threshold of mean normalized Malthusian growth. Figure 2b shows the number of cycles of fungicide exposure gained or lost to reach critical values of $M_{it}^n$ compared to the lines continuously exposed to the same fungicide, depending on alternation components. Alternation decreased the resistance evolution rate of the high-risk fungicide mostly towards the end of the selection phase and during the established-resistance phase, rather than during the early phases (delaying the established-resistance phase by 0.77 and 1.31 cycles for alternations of two and three AIs, respectively). For the medium- and low-risk fungicides, this beneficial delaying of the established-resistance phase was observed only in the case of alternation with the partner of lowest possible risk (delayed by 1.70 and 1.01 exposure cycles, respectively), with the high-risk partner decreasing the time required to reach the established-resistance phase (by 1.22 and 2.42 exposure cycles, respectively). By contrast, alternation with a partner with a higher resistance risk generally increased the rate of selection of resistance to the low-risk fungicide, particularly during the emergence and early selection phases (by 1.52–2.13 exposure cycles).

The relative selection pressure exerted by each fungicide during alternation, therefore, varied not only according to the selection regime and its components but also according to the phase of resistance dynamics. This contrasting pattern of selection over time can be first explained by the decreasing efficacy of the alternation partner, as resistance spreads more or less rapidly depending on its resistance risk. Indeed, a low-risk alternation partner may control population size longer (i.e. possibly until the latest phases of resistance dynamics) than a high-risk one, which leads according to our protocol to greater immigration from the control lines and then to greater dilution of the selected resistance in the population initiating the upcoming selection cycles. By contrast, while losing its efficacy more quickly, a high-risk

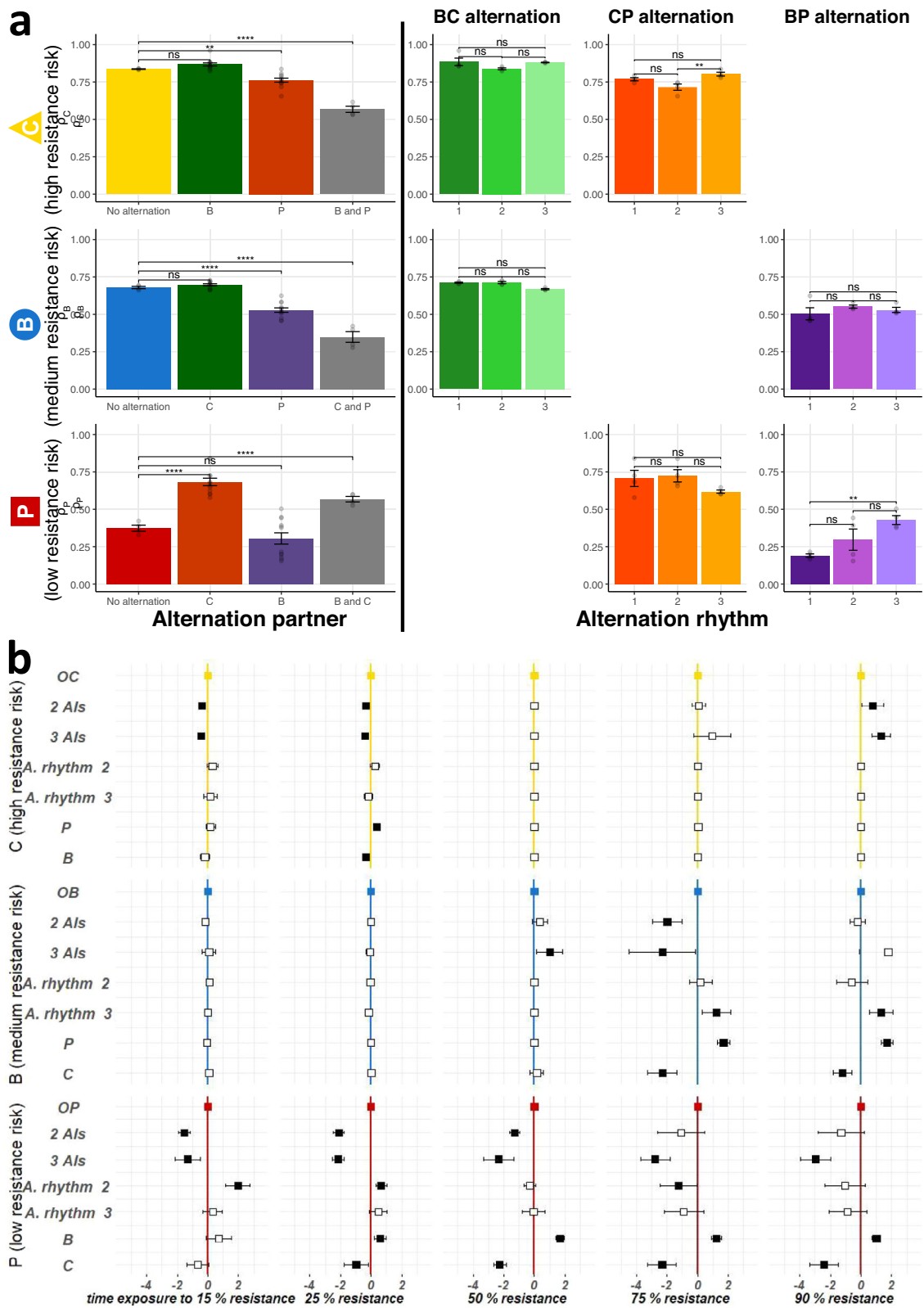

alternation partner is expected to increase resistance selection from the early phases of selection, especially since the immigration of susceptible cells from the control lines is rapidly unnecessary. The observed patterns of selection evolution are globally consistent with this rationale. However, discrepancies between observed and expected patterns may result from the exploration of adaptive landscapes[49], i.e. the successive selection

of different mutations increasing fitness over the course of the experiment, to adjust to the overall selection pressure. We, therefore, analyzed the change in population composition over time in regimes under continuous exposure (lines 0B, 0C, and 0P; Fig. 3), to obtain proof-of-concept. As expected, resistance profiles were generally narrow and mostly focused on resistance to the fungicide used for continuous selection. However,

**Fig. 2 Effects of alternation components on the evolution of resistance. a** $\rho_f$, the mean rate of resistance evolution for the time of exposure to fungicide $f$, is shown as a function of alternating AIs on the left, and for each alternation pair, as a function of alternation rhythm on the right. The error bars represent the standard error. *P*-values for pairwise comparisons were calculated with linear models (Tukey's post-hoc correction). "ns" not significant ($P > 0.05$), *$P < 0.05$, **$P < 0.01$, ***$P < 0.001$. Alternation decreased resistance evolution relative to continuous exposure, except for the low-risk fungicide (P) in alternation with the high-risk fungicide (C). Fast alternations (low rhythm) are recommended, even if the effect of rhythm alternation is often weak. **b** The effect of alternation components on Malthusian growth $M_{it}^n$ is shown as the exposure time required to reach several thresholds. These results were estimated by Cox analysis and are expressed as cycles of fungicide $f$ exposure gained or lost to reach these critical values relative to the lines continuously exposed to $f$. From top to bottom: C, B, and P. Squares represent the estimated effects and error bars at their 95% interval. Black and white squares correspond to significant and non-significant effects, respectively. Alternation had a positive effect on Malthusian growth (i.e. the time taken to reach a particular threshold was greater in lines subjected to alternating regimes), mostly during the established-resistance phase (resistance >90%), regardless of the alternation partner for the high-risk fungicide C, but depending on alternation partner for B and P. For the low-risk fungicide P, the effect of alternation on Malthusian growth was negative (i.e. alternation with partners with a higher resistance risk decreased the time taken to reach thresholds) during the early phases of an alternation.

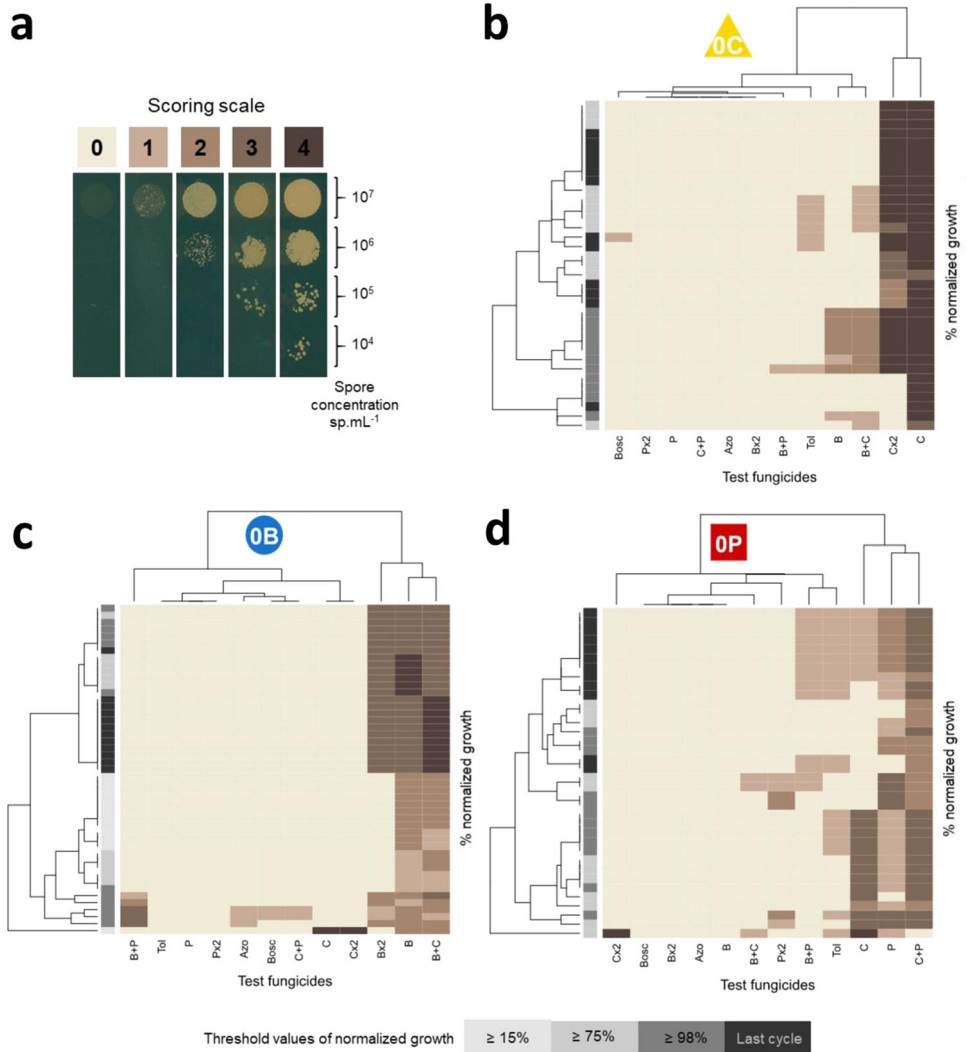

**Fig. 3 Evolution of phenotype resistance profiles during experimental evolution. a** Scoring scale was used to assess the growth of isolates on a solid medium supplemented with discriminatory doses of fungicides (droplet tests). For each isolate, four spore concentrations were tested on the same plate. 0: no growth (susceptible isolate). 1–4: growth varying with spore concentration (resistant isolate). The fungicides tested included B, C, and P at their selection doses, at twice this dose, and in mixtures, as well as fungicides not used in the experiment (boscalid, azoxystrobin, tolnaftate). Tolnaftate was used to identify isolates displaying multidrug resistance due to enhanced efflux. **b–d** Heatmaps of phenotype resistance profiles scored for isolates collected during experimental evolution, and for lines continuously exposed to C, B, and P, respectively. For each line and fungicide test, the resistance rating scores (0–4; represented by the brown scale) of 12 isolates collected from the four replicates are shown for three to four critical values of spore concentration in lines subjected to alternating regimes relative to control lines (≥15% final frequency; represented by the gray scale). The structure of the lines under continuous B, C, or P selection revealed significant changes between these different stages of resistance evolution (AMOVA; $0.273 < F_{st} < 0.453$; $P < 0.001$).

phenotypic resistance profiles differed significantly between different resistance dynamics stages. In AMOVA analyses, within-selection regime diversity was always greater than between-selection regime diversity but was concentrated throughout evolution, in a manner that varied with the fungicide exerting the selection pressure. The high-risk fungicide (C) ultimately drove the selection of four highly specialized profiles (resistant to a mean of 1.25 AIs). By contrast, the medium- and low-risk fungicides (B and P) selected multiple resistance phenotypes, differing in terms of both cross-resistance and resistance intensity. The medium-risk fungicide selected patterns of resistance to a mean of 2 AIs. The patterns selected by the low-risk fungicide were the most generalist (resistance to a mean of 4 AIs) and often included resistance to tolnaftate, an indicator of multidrug resistance in several pathogens[43]. In alternating lines, such contrasting patterns of adaptation to a single AI should recur regularly as AIs succeed each other over the generations, thereby resulting in even more complex trajectories.

**Fungicide alternation preferentially selected for multiple or generalist resistance mechanisms**. The effect of the anticipated greater selection heterogeneity of alternation regimes was investigated at the end of the experiment, by establishing the phenotypic resistance profile of the evolved individuals in each line. Therefore, we analyzed the average ability of growth under 12 different testing modalities of selected isolates (Fig. 4a) and established heatmaps showing the diversity of these phenotypic resistance profiles according to the origin of the tested isolates (Fig. 4b). The profiles of strains evolving under direct selection regimes were generally narrower and less diverse than those of strains evolving under alternation regimes. PCA confirmed this finding (Fig. 4c), with direct selection clusters being more concentrated than any other and well separated from those displaying resistance to fungicides not used for selection (e.g. boscalid, another SDHI from a different chemical class and azoxystrobin and tolnaftate, with different modes of action). The diversity of the lines was further explained by resistance to the low-risk fungicide (P), which structured the second axis of the PCA. The width of multiple or cross-resistance increased significantly with the number of AIs used in the selection regime and depended, to a lesser extent, on the alternation partner. Resistance to tolnaftate, reflecting multidrug resistance, was significantly more frequent in lines treated with alternating fungicide regimes with a long alternation rhythm. The intensity of resistance was significantly decreased by alternations including medium- and low-risk fungicides but increased slightly with an increasing number of AIs in the selection regime and with long alternation rhythms.

The broader resistance profiles of strains subjected to alternation regimes are potentially consistent with the selection of multiple resistance (i.e. the co-selection, in the same strain, of several mutations, each conferring resistance to a single fungicide) and/or generalist resistance (i.e. the selection of pleiotropic mutations, associated with a more general resistance to fungicides). We initially tested these hypotheses by sequencing the genes encoding the targets of carbendazim, prothioconazole, and benzovindiflupyr in 141 isolates corresponding to the various phenotypic resistance profiles. Only the E198K substitution in β-tubulin was detected in a few isolates evolving under direct carbendazim selection. This target site mutation has been reported to confer a high degree of resistance to benzimidazoles in field isolates of 12 fungal species, and in nematodes[50,51]. No other change was recorded relative to the 330 full-gene sequences established for *tub2, cyp51, sdhB, sdhC,* or *sdhD*. The systematic genotyping of the promoter regions of *tub2, sdhC,* and *cyp51* did not reveal any indels for these strains by comparison to the

ancestral isolate. The normalized expression of these genes, quantified by qRT-PCR after cultivation without fungicide induction, varied roughly between 0.5 and 4 comparatively to that of the ancestral isolates (Supplementary Fig. S2). As these weak variations included a susceptible isolate collected from the control line, we assumed that overexpression of these three target genes did not operate in the dataset or was rather weak. Enhanced efflux is a non-target site resistance mechanism associated with low RFs per se but causing cross-resistance to unrelated compounds in *Z. tritici*[43]. In field strains, it is determined by insertions in the promoter region of the MFS1 transporter gene[52]. Systematic genotyping of 249 isolates collected in this experiment revealed no changes in the length of the *mfs1* promoter, particularly in strains displaying tolnaftate resistance. However, qRT-PCR revealed approximately 5-, 11- and 30-fold overexpression of *mfs1*, compared to the ancestral strain, in three out of eight isolates cultivated without fungicide induction and collected from lines subjected to alternation regimes (Supplementary Fig. S2). These isolates exhibited moderate to low scores of tolnaftate resistance. Overall, these findings suggest that alternation components favor generalist phenotypes, with enhanced efflux due to the MFS1 transporter, regulated by genomic changes differing from those occurring in the field in some isolates, occurring in some but not all isolates. The low correlation observed between scores of tolnaftate sensitivity and *mfs1* expression implies that unknown transporters are also frequently observed in this context, possibly in addition to other unknown non-target site resistance mechanisms.

## Discussion

This study confirms the value of experimental evolution as an approach for investigating conceptual issues in temporal adaptation, such as the factors affecting the adaptive differentiation of local populations[48,53,54]. Here, we used an experimental evolution approach and the economically important wheat pathogen *Z. tritici* as a model for general principles to demonstrate that temporal heterogeneity can mitigate the evolution of resistance to fungicides in lines subjected to treatment with fungicide alternation regimes. These experiments clearly highlight the relationships between the components of the temporal heterogeneity of fungicide selection and both resistance dynamics and the generalism or specialism of the evolved isolates. With this experimental design, we were able to validate, in a filamentous plant pathogen in vivo, the theoretical assumptions that temporal heterogeneity of selection can quantitatively modulate adaptation[10,17] and mediate generalism in evolved individuals[27,55], as previously demonstrated for weeds, insects, and bacteria[12,29,56–59].

Specifically, using three different fungicides, we found that alternation was either neutral or beneficial in terms of delaying the overall resistance evolution rate, relative to the continuous use of a single fungicide, whatever its resistance risk. In all cases, alternation strategies were performed at least as well as the continuous use of the lower-risk fungicide. Alternation may delay the build-up of resistance to a particular fungicide because the selection pressure exerted by the fungicide is spread over a longer period than in a sequence strategy[21]. However, this would suggest that fungicide-specific resistance evolution rates ($\rho_f$) are similar for continuous-use and alternation strategies, and independent of alternation partner and alternation rhythm. By contrast, during our dissection of the impact of alternation components, we obtained variable $\rho_f$ values and revealed the primary influence of the fungicide resistance risk, determined by its mode of action, relative to alternation partners, on selection and the overall performance of a strategy. Thus, increasing the number of AIs in the

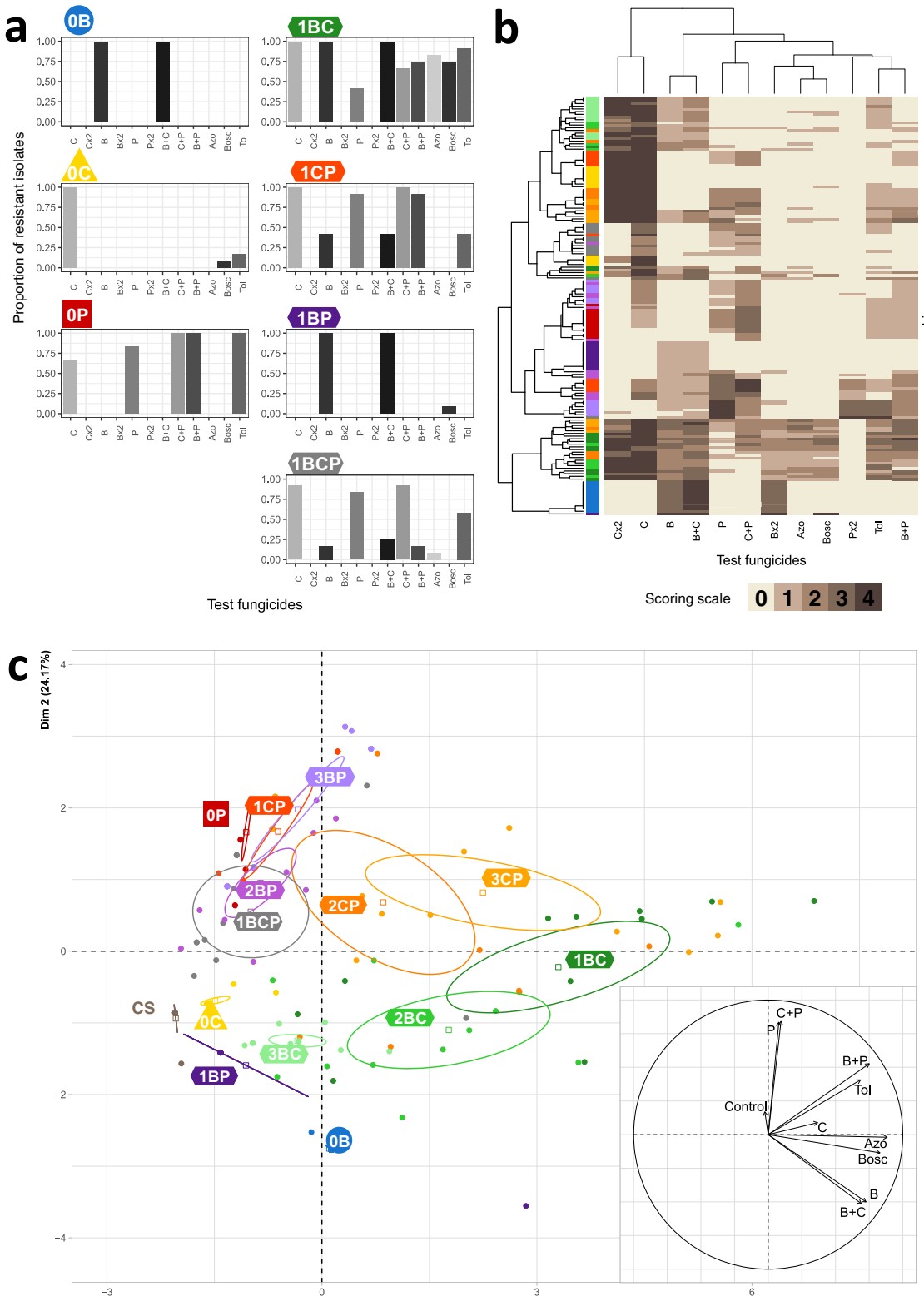

alternation may either increase or decrease the fungicide-specific resistance evolution rate, depending on the intrinsic risk of resistance for the added AI relative to those already present in the alternation. This may reflect the balance between the relative fitness costs and levels of resistance of the multiple genotypes selected by a given AI in a cycle, which finally determines the composition of the population at the beginning of the next cycle,

before contrasting selection with a different AI operation. Our observation then also reflects the relative effectiveness of each partner over time in relation to the current population composition. When needed to prevent line extinction, in the early cycles, immigration of susceptible isolates from the control lines contributed to slow down resistance but promoted the relative effectiveness of both partners. Finally, the temporal heterogeneity

**Fig. 4 Phenotypic resistance profiles selected at the end of experimental evolution. a** The average phenotypic resistance profile obtained with the different selection regimes (i.e. after continuous or alternating exposure; alternation rhythm = 1 cycle) at the end of the experiment. Diagrams show the proportion of isolates resistant (rating score > 0) to the test fungicides among the 12 collected from the four replicate lines of each selection regime. The test and scores are described in the legend to Fig. 3. **b** Heatmap of the phenotypic resistance profiles at the end of experimental evolution. The resistance rating scores (0–4; represented by the brown scale) are shown for each of the 12 isolates collected for each of the 13 selection regimes (4 replicate lines per regime; represented by the rainbow scale, as described in Fig. 1) and for each fungicide tested. The structure of the lines differed significantly between selection regimes (AMOVA; $F_{st} = 0.455$, $P < 0.001$). Alternation favored generalist phenotypes: the number of AIs to which a line is resistant (excluding tolnaftate) increased with the number of AIs used in the selection regime ($\chi^2 = 36.5$; df = 2; $P = 10^{-8}$) and was also dependent, to a lesser extent, on alternation partner. Similarly, resistance to tolnaftate, an indicator of multidrug resistance in several pathogens, increased with the number of AIs used in the selection regime ($\chi^2 = 4.91$; df = 2; $P = 8 \times 10^{-3}$) and alternation rhythm ($\chi^2 = 3.85$; df = 2; $P = 0.022$). The intensity of resistance, assessed with rating scores, was significantly affected by all components of an alternation. In particular, alternations including the medium- and low-risk fungicides decreased the intensity of resistance ($F = 70.17$; df = 1,225; $P = 6 \times 10^{-15}$ and $F = 57.06$; df = 1; $P = 10^{-12}$, respectively). By contrast, alternation rhythm and the number of AIs used in the selection regime slightly increased the intensity of resistance ($F = 4.91$; df = 2; $P = 0.008$ and $F = 3.85$; df = 2; $P = 0.02$, respectively). **c** PCA on the phenotypic resistance profiles for all lines at the end of the experiment showed structuring according to independent AIs (not used for selection during the evolution experiment; Azo and Bosc) and according to the low-risk fungicide (P). CS control solvent lines.

of environments may determine the direction of evolution, with more fine-grained temporal environments (in which the environment varies over time scales shorter than the generation time) being more efficient at delaying adaptation while maintaining population diversity[60,61]. In our design, even the one-week alternation rhythm greatly exceeded the generation time of *Z. tritici* (estimated at 6–7 generations per cycle), so all the temporal environments tested were coarse-grained, particularly the regimes with a 3-week alternation. However, alternation rhythm had a much lesser impact than other components on the resistance evolution rate of a given fungicide. This implies that some flexibility is possible in the design of field strategies, as the frequency of fungicide sprays is likely to create coarse-grained environments in many agrosystems. The respective relevance of these selection drivers for mixture strategies, most commonly used in the field, could be interestingly compared in further studies.

We confirm here, in a plant pathogenic fungus, that temporal variation in fungicide application may be correlated with a higher degree of generalism in resistant strains. An analysis of resistance phenotype profiles in the course of direct selection (continuous use of the same fungicide) revealed high within-line phenotype diversity and the concentration of this diversity over time, culminating in specialist profiles, for which the degree of specialism depended on the AI. These dynamics are also consistent with the continual change in population composition, with the rise, selection, fixation, or extinction of beneficial mutations, depending on their relative fitness costs and the degree of resistance in the local population[62]. High-risk fungicides can thus explore smooth fitness landscapes, such as those generated by the selection of a unique change underlying high resistance (e.g. the tub2 E198K variant conferring high resistance to C). By contrast, medium- and low-risk fungicides can explore less smooth landscapes. According to theory, the global fitness optimum is less likely to be achieved in variable environments[63]. Combining the respective paths associated with AIs not exhibiting cross-resistance, and considering the population dynamics as previously discussed, we would then expect lines evolving under alternation regimes to explore rugged landscapes, constraining resistance mutations to one fungicide with trade-offs in resistance to other fungicides or to favor genotypes with a higher average fitness over time[29,64]. This scenario is consistent with the fixation of multiple discrete resistance mechanisms, together with the accumulation of relative fitness costs associated with each resistant trait, as suggested by the wider patterns of cross-resistance (including resistance to new AIs), lower resistance scores, and higher frequency of resistance to tolnaftate (associated with enhanced efflux) observed in isolates collected from lines subjected to alternating treatments. The molecular analysis of a representative

subsample of resistant isolates hardly detected mutations in genes encoding the targets of selecting fungicides and indicated that enhanced efflux due to the MFS1 transporter was at work in a few —but not all—resistant strains. Our collection, therefore, provides a good opportunity to make use of the progress in genomics to elucidate unknown genetic and non-genetic bases of adaptation mostly at work in our dataset. These unknown resistance mechanisms (i.e. other than target alteration or overexpression), may also operate *in natura*, without our suspecting it, and may, therefore, have an overlooked impact on resistant traits[65–67]. Indeed, genomic analysis has shown that not only SNPs but also indels, copy number variants, transposable element insertions, chromosome duplications, and aneuploidy occur in fungal pathogens exposed to a sublethal dose of fungicides[68]. Our observation of unknown resistance mechanisms in isolates from alternation as well as sequence strategies, is consistent with this pan-genome characterization of fungal adaptation to fungicides, without a priori on resistance mechanisms. This finding brings the proof of concept that *Z. tritici* is also a relevant model to explore the various shades of non-target site adaptation, in addition, to informing on the general principles to manage resistance, whatever mutations were selected. We also found that alternation components had different impacts on the rate of resistance evolution over the course of the experiment. Alternation significantly delayed the establishment phase of the dynamics of resistance to the high-risk fungicide, suggesting that lower-risk fungicides perform better than high-risk ones after the emergence phase. By contrast, alternation with high-risk fungicides decreased the time required for the early stages of resistance to be reached for a low-risk fungicide, demonstrating that the high-risk fungicide did not counter-select mutations with a low resistance intensity. Finally, experimental evolution has proposed a simplified model for evolution *in natura*[54] and enriched our understanding of the general principles of resistance management by highlighting the relative impact of selection components. Due to the laboratory environment, our experimental design assumes a finite population size, low ancestral diversity, limited time scale, and asexual reproduction. However, many plant pathogens, including *Z. tritici*, have large population sizes, high levels of genetic diversity, and both sexual and asexual reproduction, and have to adapt simultaneously to their host and abiotic environment[69]. Gene flow is also expected to occur between evolving metapopulations. Because of these differences, we felt that selecting for this experiment adaptive mutations distinct from those occurring in the field would not limit the relevance of this study for reasoning about antifungal management strategies. Beyond these limitations, our study clearly demonstrates that not all alternation strategies guarantee the same performance for

preventing resistance, but in no instance did these strategies increase the risk of resistance relative to that associated with the continuous use of individual fungicides. It might then offer an alternative to the mixture strategy to delay resistance, while reducing pesticide load, a growing socio-environmental demand. Indeed, for example, commercial mixtures used on wheat to control Septoria leaf blotch combine fungicide components at rates close to those recommended for their individual use, resulting in the application of almost double the amount of fungicide[70]. Efforts to educate stakeholders about alternation strategies should, therefore, focus on optimizing the interplay between the various components of an alternation. The results of this study pave the way for the effective tailoring of resistance management for plant pathogens based on temporal heterogeneity, especially in perennial crops where multiple sprays are often applied yearly. We would also argue that resistance management via temporal heterogeneity is possible provided that the environmental grain is appropriate[61]. The environment should generally be fine-grained, requiring alternating generations to be exposed to compounds with a different mode of action, but we found that coarse-grained temporal fluctuations were potentially acceptable. Furthermore, the order in which the AIs are cycled may affect the trajectory of evolution. Modeling suggests that the most effective AI should be used first[21], to control emerging mutations as efficiently as possible, but further explorations in experimental evolution experiments would be required to confirm this. The overall performance of alternation approaches including AIs with different inherent resistance risks is achieved at the expense of the AIs with the lowest risk of resistance, suggesting that the use of high-risk AIs in alternation sequences should be limited and/or that the alternation of AIs of similar risk should be preferred. The relative risk between AIs might also be adjusted in the field by using different doses or alternation rhythms for each partner, or by mixing some of the alternated AIs. The efficacy of alternation strategies, as well as of mixtures, should also be confronted in the real-world case in which populations are not naïve and the initial levels of resistance to each fungicide differ, not to mention multiple resistance[71]. Finally, the design of rational strategies with optimized components should consider the evolutionary trade-off between the rate of resistance evolution and the degree of generalism of the evolved individuals, which may call into question the efficacy of future modes of action. In this respect, the future control of weeds, pests, and diseases should be based not only on a diversity of modes of action but should consider integrated pest management including non-chemical strategies to mitigate overall adaptation[72].

## Methods

**Ancestral population, culture conditions, and assessment.** We used IPO-323, as an ancestral isolate, to find all lines, i.e. similar ancestral populations further evolving under fungicide selection. This isolate is susceptible to all fungicides and was the source of the high-quality reference genome for *Z. tritici*[73]. The stock spore suspension was kept at $-80\,^{\circ}C$ in 20% glycerol and gently thawed at room temperature for the inoculation of solid YPD plates (20 g L$^{-1}$ dextrose, 20 g L$^{-1}$ peptone, 10 g L$^{-1}$ yeast extract, 20 g L$^{-1}$ agar; USBiological), which were then incubated at $18\,^{\circ}C$ in the dark for 7 days. These plates were used to prepare precultures in liquid YPD (as previously described but without agar), which were shaken at 150 rpm in similar conditions for seven days before the start of the experiment.

The culture medium used for experimental evolution was liquid YPD supplemented with 100 mg L$^{-1}$ streptomycin and penicillin to prevent contamination of this long-term experiment. The culture vessels were 50 mL borosilicate Erlenmeyer flasks containing 25 mL medium, with carded cotton wool inserted in the neck. Cultures were grown in the dark at $18\,^{\circ}C$, with an RH of 70% and shaking at 150 rpm during the duration of the cycle. In these conditions, *Z. tritici* can be kept in its yeast-like form, making it possible to obtain homogeneous liquid cultures[74].

Population size was routinely determined by measuring optical density at 405 nm (or OD$_{405}$) for two 200 μL replicates of the fungal culture in a 96-well microtiter plate (Sarstedt) sealed with a gas-permeable membrane (Breathe Easy®;

Diversified Biotech). The detection threshold of the spectrophotometer (SpectraMax M2, Molecular Devices) was $3.5 \times 10^5$ spores mL$^{-1}$ in the experimental conditions used. The OD$_{405}$ of the fungal cultures was normalized against that of the non-amended medium. Spore concentration was estimated from the OD$_{405}$ with a mathematical model calibrated on the basis of preliminary experiments in our conditions with serial dilutions and hemocytometer-based spore quantification, and cross-validated ($R^2 = 0.91$). Population size in spores mL$^{-1}$ was used in further statistical analyses.

**Fungicides and selection regimes.** In the experimental evolution experiments, we selected resistance to three fungicides representative of different modes of action: carbendazim (C; benzimidazole, interfering with β-tubulin assembly in mitosis), benzovindiflupyr (B; pyrazole carboxamide, inhibiting complex II, succinate dehydrogenase, in the mitochondrial respiratory chain) and prothioconazole-desthio (P; triazole, inhibiting the sterol 14α-demethylase during sterol biosynthesis), the active metabolite of prothioconazole. These modes of action are associated with contrasting resistance risks. The fungicides were dissolved in 80% ethanol. The concentration of this solvent never exceeded 0.5% of the final culture volumes.

The selection doses were chosen by examination of dose–response curves for the ancestral strain. For each fungicide dose, $10^7$ spores of IPO-323 were used to inoculate YPD in an Erlenmeyer flask, which was then incubated for 7 days before OD$_{405}$ measurement. OD$_{405}$ data were modeled by logistic regression, to calculate, for each fungicide, the EC$_{50}$ (concentration inhibiting 50% of growth), EC$_{95}$ (concentration inhibiting 95% of growth), and minimum inhibitory concentration (MIC) values. We finally chose to use the EC$_{95}$ as the selection dose, as 99% inhibition was not achieved for some fungicides, for which the MIC would exceed fungicide solubility in the solvent. The selection doses used were 0.5 mg L$^{-1}$ for benzovindiflupyr, 0.1 mg L$^{-1}$ for carbendazim, and 0.005 mg L$^{-1}$ for prothioconazole-desthio.

Selection regimes were organized for analysis of the respective impacts of three components of alternation on resistance evolution, as described in Fig. 1b. First, in treated lines, the number of AIs in the alternation regime ranged from one (direct selection) to three different fungicides alternated over time. Second, the intrinsic risk of resistance differed between MoAs and was predicted to be high for benzimidazoles (represented by C), moderate to high for SDHIs (represented by B), and moderate for DMIs (represented by P) by agrochemical companies[75]. In practice, as we observed similar relative ranking of these fungicides in our experiments, we referred to high-, medium-, and low-risk fungicides, respectively, throughout this work, for the sake of simplicity. Finally, the duration of exposure to the same fungicide (or alternation rhythm) was continuous for direct selection (0, no alternation) and ranged from 1 (fungicide changed every cycle) to 3 (fungicide changed every three cycles) cycles. All combinations of these three possible components were tested, except for the three-fungicide alternation, which was tested only for an alternation rhythm of one cycle.

**Design of the experimental evolution experiment.** The ancestral population was founded from the 7-day liquid YPD preculture of IPO-323. The density of this spore suspension was adjusted to $2 \times 10^7$ spores mL$^{-1}$ and each Erlenmeyer flask was inoculated with $10^7$ spores (500 μL). The resulting starting concentration of $4 \times 10^5$ spores mL$^{-1}$ offered a compromise between large populations in which mutations were likely to occur and not exceeding the culture carrying capacity in the experimental conditions used, based on the findings of preliminary experiments. Each of the 14 selection regimes (1 control and 13 different treatments) was repeated four times (four lines per selection regime). Control solvent lines received only 0.5% solvent, whereas treated lines were treated as shown in Fig. 1b, giving rise to 56 independent lines. Control lines without fungicides nor solvent were grown only to provide susceptible cells for immigration (see below) and were not included in the statistical analysis comparing strategies.

Experimental evolution was conducted over 12 cycles of 7 days each (about six to seven generations per cycle). This cycle duration corresponds approximately to the time for which a control line remained in the exponential growth phase before reaching a plateau. OD$_{405}$ was measured at the end of each cycle and transformed into population size, as described above. The OD$_{405}$ of the treated lines was normalized with the mean OD$_{405}$ of the control solvent lines. At each transfer, 500 μL of the evolving culture (i.e. 2% of the total volume of medium) was transferred to a new Erlenmeyer flask containing fresh medium. If the number of spores in the 500 μL of culture medium was <$10^7$, which would be the case before the evolution of resistance, then the appropriate number of spores from one of the untreated populations was added to make the number of spores present up to $10^7$. The starting population size was equalized between the lines at the beginning of each new cycle, and the process mimicked the immigration occurring in field situations. For each of the four replicates, the same source population was used for immigration throughout the experiment. In preliminary experiments, immigration was found to be useful for preventing population extinction after weekly bottlenecks and to minimize genetic drift.

At the end of each cycle, 2 mL of each line was mixed with glycerol (25%), frozen, and stored at $-80\,^{\circ}C$ for further analyses.

**Statistical analysis of resistance evolution rate**. The Malthusian growth[76] of each line was calculated as

$$M_{it} = \ln\left(\frac{\text{Spore concentration of line } i \text{ at the end of cycle } t}{\text{Spore concentration of line } i \text{ at the beginning of cycle } t}\right) \quad (1)$$

and normalized against that of the solvent control line $C$ such that

$$M_{it}^n = \frac{M_{it}}{M_{Ct}} \quad (2)$$

For each selection regime, the mean Malthusian growth ("overall resistance evolution rate" or $\rho$) is the mean of $M_{it}^n$ calculated over the 12 cycles and constitutes a quantitative summary of the increase in resistance of the four replicates over the entire experiment. Similarly, $\rho_f$ ("fungicide-specific resistance evolution rate") reflects the increase in resistance associated with a fungicide f during the alternation regime, taking into account only the time segments during which this fungicide was used. This approach made it possible to compare selection due to the same number of fungicide applications (6 applications, except for the BCP regime for which there were only 4 applications), but in the context of different selection regimes. $\rho$ and $\rho_f$ were explained in ANOVA linear models, initially considering the alternation partners as qualitative fixed factors, and then the alternation rhythm for each alternation partner. The reference used was the continuous application of the fungicide.

The exposure time (i.e. the number of cycles exposed to a given fungicide) required for the normalized Malthusian growth $M_{it}^n$ to reach certain thresholds (15%, 25%, 50%, 75%, 90%) was investigated in a Cox survival analysis taking into account the number of AIs, the alternation partner and the alternation rhythm.

All analyses and figures were produced with R 4.0.4 and the additional packages listed in Supplementary Methods S1.

**Isolation and phenotyping of evolved individuals**. We assessed the change in resistance phenotypes during the course of selection, by isolating pure strains from direct selection regimes, at critical points in the resistance dynamics, i.e. when normalized Malthusian growth reached 15% (early resistance), 75% (intermediate resistance frequency) and 98% (established resistance) relative to that of the control line, and at the end of the experiment. We isolated 12 individuals (3 per replicate) per selection regime from cycles 1, 5, 6, and 12 for 0B lines, from cycles 2, 3, and 12 for 0C lines, and from cycles 5, 6, and 12 for 0P lines. This resulted in the isolation of 120 strains in total. We also explored phenotype diversity within and between selection regimes, by systematically isolating 12 strains (3 per replicate) at the end of the final cycle, for all selection regimes. This resulted in the isolation of 129 additional strains. Isolates were retrieved from glycerol population stocks collected at the end of each cycle. Two successive isolations were performed on YPD solid medium supplemented with 100 mg L⁻¹ streptomycin and penicillin. Single isolates were stored at −80 °C in 25% glycerol.

The cross-resistance patterns of the 249 isolates were established in droplet tests. We deposited $10^7$, $10^6$, $10^5$, and $10^4$ spores mL⁻¹ suspensions as drops organized into columns, in 12 cm square Petri dishes. These plates contained YPD supplemented with discriminatory doses of fungicides. The 15 treatments tested encompassed a control with 0.5% solvent and fungicides B, C, and P at their selection doses (0.1, 0.5, and 0.005 mg.L⁻¹, respectively) and twice their selection doses (0.2, 1, and 0.01 mg L⁻¹, respectively). Mixtures of pairs of the three fungicides were also included (each fungicide used at 0.8 times its selection dose, as this was found to be the lowest discriminatory dose in mixtures of the reference resistant and susceptible strains). Finally, for the detection of generalist resistance, we also included fungicides not used for experimental evolution (0.5 mg L⁻¹ azoxystrobin, 2 mg L⁻¹ boscalid, and 2 mg L⁻¹ tolnaftate). Tolnaftate, in particular, identified individuals displaying enhanced fungicide efflux leading to non-specific multidrug resistance (MDR), as described in field isolates of *Z. tritici*[43]. The ancestral strain IPO-323, and three field strains, each resistant to one or several of the fungicides used for selection, were included as control strains. After 7 days of incubation, drops were scored 0 or 1, according to the presence or absence of fungal growth. A total score was established for each strain, ranging between 0 (susceptible) and 4 (resistant if >0), according to the number of dilution droplets for which growth was observed. For each strain, the resistance phenotype profile was determined as the concatenation of scores established for the 14 sets of fungicide conditions. In total, 132 different resistance phenotype profiles were identified.

**Statistical analysis of resistance phenotype profiles**. The evolution of strains isolated from the 0C, 0B, and 0P direct selection lines over the course of the experiment was represented by heatmaps of 13–17 resistance phenotype profiles detected during three to four critical time windows in resistance dynamics. The Euclidean pairwise distance was used for the hierarchical clustering of these profiles, with dendrograms for the rows and columns. The heatmap of the 102 resistance phenotype profiles detected at the end of the experiment in the 56 lines was produced in the same way. We also performed principal component analysis (PCA) on the same data subsets.

Diversity within and between selection regimes, measured at the end of the experiment, was quantified by an analysis of molecular variance (AMOVA) performed with the Arlequin program[77]. AMOVA is generally used for molecular

data. Here, we assumed that different scores for a particular test corresponded to different alleles or combinations of alleles.

Finally, the number of AIs to which a line is resistant and the intensity of resistance (i.e. in each test and for each line, the mean resistance score calculated from individual droplet scores >0) were calculated from the resistance phenotype profiles recorded at the end of the experiment. The effects of AI number, alternation rhythm, alternation partner (and their interaction) on resistance to tolnaftate, and the number of AIs to which a line was resistant and the intensity of resistance were investigated with linear models (quasiPoisson GLM model, Poisson GLM, and log-transformed linear models, respectively).

All analyses and figures were produced with R 4.0.4 and the packages listed in Supplementary Methods S1.

**Sequencing, genotyping, and gene expression of evolved isolates**. Target site resistance to B, C, and P has been described in field strains of *Z. tritici* and is associated with mutations of genes encoding some of the subunits of succinate dehydrogenase (*sdhB*, *sdhC*, or *sdhD*), β-tubulin (*tub2*) and 14α-demethylase (*cyp51*), respectively[34,39–41]. In addition to the reference isolates used in droplet tests, isolates from treated populations with scores for B, C, or P resistance in the droplet test was >1 were retained for target gene sequencing. The target genes of the fungicides concerned were amplified with primers, under the PCR conditions detailed in Supplementary Methods S2, and were sequenced with the Sanger protocol (by Eurofins Genomics, Ebersberg, Germany), for single isolates representative of their resistance phenotype profile, chosen at random. DNA was extracted from 141 individuals, and 65 *sdhB*, 69 *sdhC*, 66 *sdhD*, 83 *tub2*, and 47 *cyp51* sequences were obtained.

In order to identify possible overexpression of these target genes, their promoter regions were amplified by PCR for the same isolates, searching for indels (Supplementary Methods S3). Insertions into the promoter of the gene *mfs1* have been shown to cause MDR in field isolates of *Z. tritici*. Amplicon size after PCR for this gene was systematically determined for the 249 isolates, to check for any MFS1 promoter alteration, with previously described protocols and reference strains[52]. At last, eight isolates exhibiting high resistance scores to B, C, P, and/or tolnaftate were chosen among the collection to represent specialist or generalist phenotypes. Their RNA was isolated at the same time as that of the ancestral isolate after a 6-day culture, as previously described, and without fungicide exposure. *sdhC*, *tub2*, *cyp51*, and *mfs1* expression was quantified for these isolates by qRT-PCR using *EF1α* and *UBC* as housekeeping genes for expression normalization of *tub2*, *cyp51* and *mfs1* and *tub2* as a housekeeping gene for the normalization of *sdhC* expression (Supplementary Methods S4). The normalized expression of these genes was shown as $2^{-\Delta\Delta Ct}$.

**Reporting summary**. Further information on research design is available in the Nature Portfolio Reporting Summary linked to this article.

## Data availability

Data were deposited on the Dryad repository under https://doi.org/10.1101/2021.07.11.451819. All other data are available from the corresponding author on reasonable request.

## Code availability

Scripts were deposited on Zenodo under accession https://doi.org/10.5281/zenodo.5106475.

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

## Acknowledgements

We thank Fabrice Blanc for facilitating the administrative organization of this Ph.D. studentship. We would also like to thank Dr. Stefano Torriani, Dr. Stephanie Bedhomme, and Dr. Mato Lagator for their sound comments about our findings. We are very grateful to Alicia Noly for her help with the analysis of the expression of target genes in evolved isolates of *Z. tritici*. A.B. was supported by a Ph.D. studentship funded by the French Ministry of Higher Education, Research and Innovation, and Syngenta France, through the CIFRE program, supervised by the National Association for Research and Technology (ANRT). This work was supported by the Plant Health division of INRAE through the STRATAGEME project.

## Author contributions

A.-S.W. and F.C. conceived and designed the study, with contributions from A.D. and A.B. A.B. performed the experimental evolution experiment. P.D. and C.D. isolated evolved strains and established their phenotypes and genotypes. A.B. and F.C. performed the statistical analysis, with contributions from A.-S.W. and A.D. The paper was written by A.-S.W. and A.B., with additional contributions from all authors.

## Competing interests

The authors declare no competing interests.
