## [Peer Review File · Communications Biology]

Reviewers' comments:

Reviewer #1 (Remarks to the Author):

The rise of fungicide resistance is a major concern for food security and one of the major strategies adopted by the industry is the use of mixed fungicide regimes - whether as tank mixes or by 'alternations' (not often really alternations in the field but really just not using the same fungicide time after time).

This strategy is not threatened by the rise of strains with resistance to multiple fungicides.

This paper uses an in vitro strategy to investigate whether alternations really do delay resistance. The in vitro approach is attractive but always means that somewhat arbitrary choices must be made which may or may not reflect the situation in the field.

This very elegant and nicely written paper describes an elaborate experiment using SEPTTR and three fungicides in repeated solo selections, in alternations, with different rhythms and a triple regime.

The results support the use of alternations. The headline result is that alternations are slower or as fast as solo applications. So far so good. The problem with the paper comes in lines 220-231. The mode of resistance in the field to the three fungicides used is well known. Target site mutations for all three, plus target site overexpression and MDR for DMI. There are some reports of target gene duplications. However in this study, no genotypes associated with the field-relevant mutations were found. (There is a mention of 'E198K in a few isolates' strains - line 220). This leaves open the possibility that the phenotypes are due to known target site gene overexpression or to a generalised upregulation of relevant genes. I would suggest that it would be good to look for this either by promoter insertions in the target genes or by transcriptomics in a few strains.

The use of fungicide mixtures has led to evolution of strains resistant to all used fungicides. Fungi evolve to grow faster in the conditions presented to them. In this study, the fungi were asked to grow at the EC95. Surprisingly, almost no strains could grow at twice these concentrations (Fig 4a). The RFs of the fungicides known to be 10+ for DMIs, 20+ for SDHIs and 100+ for MBCs. Hence the phenotype of the strains found in this paper are not the same as the ones typically found in the field (I concede that the area of resistance with very small RFs is understudied).

Minor points.

The title - As I read the data, the outlier was the MBC. This is described in the paper as the high-risk fungicide. So the title could be 'Delayed but generalist resistance in alternation but only for low and medium risk fungicides.'

The abstract and intro seek to generalise the results to all pesticides and even clinical antibiotics. I would suggest limiting the scope to agricultural fungicide. My discussions with herbicide colleagues suggest that the detoxification MORs present a totally different picture of resistance management. Line 73 - the authors distinguish TSR (target site resistance) and NSTR. I would suggest that TSR is two phenomena - mutations in the coding region (TSM) and overexpression (TSO). Furthermore there can be many TSMs in a single strain.

331 - why use 100 mg/ml strep and pen?

These are high glucose low oxygen conditions

The paper is hard to read at times. The figures are packed and fonts are small. The three fungicides are C, B and P and are high, med and low risk but in Fig v3 are in panel B, C and D. I'd suggest using the acronyms CBX, PTZ and BZF in figures and the text.

119 and 163 - the sigmoid rise of resistance to PTZ and BZF. Why would this be? I am not sure this is equivalent to the emergence and selection phase.

168 Fig 2c is missing

166 and 201 - the reader would be helped by a sentence or two describing the figure. Much of this text is in the figure legend but it should instead/also be in the text.

Reviewer #2 (Remarks to the Author):

This paper uses experimental evolution to test fungicide resistance evolution under different management strategies; specifically, the alternation of different chemical modes of action. The authors clearly explain that their findings will be relevant to any cases of resistance evolution and are therefore of wider interest beyond fungicides and plant pathology.

In all cases, resistance was selected more slowly under alternating fungicides of different modes of action than with repeated use of a single compound. However, the rate of resistance evolution per treatment with a given compound was only reduced in alternations with fungicides of lower resistance risk.

They also observe greater cross-resistance between different fungicides in lines selected with alternating fungicides, compared to more specialised resistance selected with a single mode of action; interestingly this extends to additional modes of action not used within the alternation.

This paper presents some interesting data, although in some cases the interpretation is somewhat speculative and not all alternative explanations have been considered. In particular, where the effectiveness depends on the resistance risk of the alternation partner, and where selection varies over time as the populations evolve, this will be partly because as resistance evolves against the alternation partner, treatments with the alternation partner fungicide will result in smaller-to-no reductions in microbial population size; and in this experimental system, this will alter the proportion of selected versus 'immigrating' population going into the next round of selection with the focal fungicide (with immigration diluting the effects of previous rounds of selection). The effects of basic population dynamics need to be accounted for before invoking hypotheses about adaptive landscapes. It should also be noted that, with the exception of Beta tubulin E198K, the evolving lines did not evolve any of the specific resistance mechanisms that have been reported in field populations. Whilst this is not a fatal flaw in a paper using *Z. tritici* as a model for general principles, how the various findings may or may not relate to field conditions should be discussed. For example, mixtures/alternations are widely used and yet target site specific resistance is common.

Specific comments

Abstract:

line 28-29: The relative intrinsic resistance risk determines the relative benefit of alternation for each fungicide; I'm not sure they demonstrate a relationship between the relative resistance risk of components and the level of generalism in the evolved lines?

Introduction:

The introduction gives a broad overview of wider questions but would benefit in places from a little more detail and some specific examples

58: What are some key areas of disagreement between models and empirical data? Which biological traits could affect the results?

67: phenotypic impact?

73: levels of cross resistance for TSR vary between mutations/ modes of action

83: "any given strategy": examples?

96: explain what strategies and how they impacted loss of efficacy?

97: the cited references cover several European countries, not just France

99: explain "not associated to TSR": TSR and MDR can both be found in the same isolates

104: define "alternation rhythm" at the first use of the term

105: what do you mean by "interplay"? patterns of cross resistance? relative resistance risk of each? Relative resistance risk of the alternation partner affected the rate of resistance evolution but in what ways did it alter the nature of the trade-off with generalism?

106: qualitative and quantitative: does this mean speed versus generalism of resistance?

Results:

The authors should further clarify, ideally within the section headings, the distinction between total time to resistance (for which alternations were always neutral to beneficial) and rate of selection during treatment with a given fungicide (for which only alternation with lower resistance risk fungicides was beneficial), otherwise the stated findings appear contradictory. See also abstract line 27: "slows the evolution of resistance" could be misleading.

121: define "generalised" at the first use of the term (especially given potential confusion between generalisation and generalism)

122: in which conditions did resistance never generalise?

130: decrease in selection compared to the lower or higher risk fungicide alone?

140: "multi-directional selection": in this case do you mean selection per treatment with the fungicide in question?

151: this is a key point and should be further highlighted: do alternations have a net benefit on longevity of the total set of fungicides, or is it simply a zero-sum combination of resistance selection for the different modes of action?

162 and following section: Are these changes over time all attributable to the adaptive landscape of the fungicide in question, or does the effect of alternation change as resistance also evolves against the alternating partner? ie. in the early generations, the alternating partner is highly effective at reducing pathogen population size, necessitating a larger input via immigration from the non-selected line. In the later generations, especially for a higher-risk mixing partner, treatment with that fungicide has little effect and is more like a treatment gap in which the population is left to grow.

182: do isolates from different time points show sufficiently distinct resistance profiles to suggest that different genotypes were selected over time? If so, please add figures demonstrating this.

220: can beta-tubulin genotypes be tracked over time (re. previous point)?

235: why specifically "local" populations?

237: again, is there a net benefit of heterogeneity here, or only zero-sum sharing of resistance selection across alternation components?

246: clarify: total time to resistance, not selection rate during treatment

268: There should also be some discussion of alternations versus mixtures

280-283: clarify whether you are referring to fungicides with negative resistance or just with no cross-resistance. I think you mean the trade-off between high resistance to a single MOA versus lower-level resistance to multiple MOAs. The point here is not necessarily that the fitness landscape under selection by any one fungicide is more rugged, but that the landscape shifts between one fungicide and another, favouring genotypes with a higher average fitness over time.

285: What is the evidence for additivity or synergy? The wider cross-resistance is an example of pleiotropy not epistasis.

296-300: Is this due to fitness costs or resistance levels of resistance mutations against the fungicide in question, or is it confounded by the relative effectiveness of the alternation partner over time? A higher risk fungicide will have more effect during the early stages, before resistance to the higher-risk fungicide evolves and it stops effectively reducing the pathogen populations; whereas a lower-risk mixing partner will continue to delay resistance even once resistance to the high-risk fungicide is in the final stages of selection.

308: When discussing the relevance to growers, again there needs to be discussion of using multiple MOAs in alternations versus mixtures.

322: Can relative risk be adjusted by using different doses or frequencies of each partner, or using some alternation partners in mixtures?

Methods

376: the predicted risk levels for each fungicide group are also supported by resistance evolution in the field

393-394: Are "control lines" and "control solvent lines" the same or were there control lines with and without solvent?

409: was the rate of genetic drift tested?

476: Need to discuss the assumption that different phenotype scores correspond to different alleles: could there be polygenic resistance? Multiple alleles with similar phenotypes?

Figures

1b: Is the triangle illustration needed? It may be clearer just to use a simple legend for the 3 individual AIs

Reviewer #3 (Remarks to the Author):

General comment

Short: An excellent piece of work put together. I do not have many comments. My main critic is that the introduction part – in contrast to the rest of the manuscript – is hard to read and needs to be rephrased before the manuscript should be accepted for publication.

Title: Consider rephrasing the title to give a clear idea on your findings, e.g., 'Alternation of antifungal modes of action confer general fungicide resistance.

The introduction is hard to read, especially the first part. Please rephrase the first paragraph with shorter sentences and be more specific when using terms such as resistance (fungicide resistance), strategies (spray strategies; L 50 "Strategies to delay fungicide resistance...").

I see the point in starting from the more general point of pesticide/drug resistance and the theories around different usage pattern selecting differently. However, very much information is described very condensely and makes is hard understand. I would suggest sticking to pesticides used in agriculture, zooming in on the *Z. tritici* model, and finishing off that these results can be transferred to human and animal health.

Give a short introduction why *Z. tritici* is a good model to assess fungicide resistance evolution.

L 38: Human societies = humanity?

L 82: "..., for which increasing use of different MoA/herbicides"

L 96: "...mutations and enhanced efflux; the latter being a generalist mechanism causing..."

L 98: yeast.like form = blastospores?

Drop the term "agents", and introduce AI sooner. Introduce and use the abbreviation MoA already in the introduction.

Results:

L 115: 'overall' instead of 'global'

L 121: "... 1.4 times higher with C than with P; these significant differences reflect ..."

L 123: "The overall resistance rate..."

L 162: Isn't 90% a high value. I would say that resistance already is generalized with 75%

L 242: "...in terms of delaying overall fungicide resistance..."

Methods

L 325: Headline to general: what ancestral population, what kind of assessment?

L 328: Is taking the isolate out of – 80C to let it thaw at room temp gentle?

L 336: delete the second "RH"

L 350: Why was carbendazim used?

L 354: associated with...

L 358: OD 405

L 382: 4×10^5 sp. mL⁻¹

L 386: "...were not treated with any fungicide"

L 407: Reference for the Malthusian model?

L 429: Isn't 75% already a high % for intermediate resistance frequency?

L 430: Personally, I am not a fan of the word "generalization" in this context. How about 'established resistance' or 'full resistance'.

L 440: "... 10^5 , and ..:"

L 453-454: You had four droplets on each AI – if growth was observed, it was scored 1. If all dropets grew, it got the soucre 4. Is that correct?

L 492-494: Why not from all isolates?

Figures:

The figures are very illustrative and helpful to understand the experimental design.

Ref.:

L 695: Is this the right way to cite a book

Answers to reviewers

>We do thank the three anonymous reviewers for their time, interest in our work and valuable comments. We carefully addressed most of them and achieved additional experiments to characterize further the resistance mechanisms occurring in evolved strains (Methods S2 and S3 and Figure S2). We do hope that the manuscript is now acceptable for publication.

Reviewer #1:

The rise of fungicide resistance is a major concern for food security and one of the major strategies adopted by the industry is the use of mixed fungicide regimes - whether as tank mixes or by 'alternations' (not often really alternations in the field but really just not using the same fungicide time after time). This strategy is not threatened by the rise of strains with resistance to multiple fungicides.

This paper uses an *in vitro* strategy to investigate whether alternations really do delay resistance. The *in vitro* approach is attractive but always means that somewhat arbitrary choices must be made which may or may not reflect the situation in the field.

This very elegant and nicely written paper describes an elaborate experiment using SEPTTR and three fungicides in repeated solo selections, in alternations, with different rhythms and a triple regime. The results support the use of alternations. The headline result is that alternations are slower or as fast as solo applications. So far so good. The problem with the paper comes in lines 220-231. The mode of resistance in the field to the three fungicides used is well known. Target site mutations for all three, plus target site overexpression and MDR for DMI. There are some reports of target gene duplications. However, in this study, no genotypes associated with the field-relevant mutations were found. (There is a mention of 'E198K in a few isolates' strains - line 220). This leaves open the possibility that the phenotypes are due to known target site gene overexpression or to a generalised upregulation of relevant genes. I would suggest that it would be good to look for this either by promoter insertions in the target genes or by transcriptomics in a few strains.

>This comment was addressed. Please refer to our answer in the editor section. This is not because field mutations were not selected that the determinants of the selection that we highlighted were not relevant. For a pathogen, adapting to fungicides in an *in vitro* environment may imply to face to different trade-offs (and especially including survival in a plant) which may explain why the genomic changes finally selected may differ. This is one of the limitations of an *in vitro* experimental design. But the mutations that we have selected might contribute to the phenotype of resistance in the field as well and the resistance phenotype dynamics might be similar. Genomic data, both in the agriculture and medical world, bring evidence that not only target site (upregulation) mutations are selected in fungal genomes. A recent GWAS analysis on resistant field isolates from our group, to be published shortly, highlights 11 new loci contributing to resistance to DMIs and SDHIs, to different extents.

The use of fungicide mixtures has led to evolution of strains resistance to all used fungicides. Fungi evolve to grow faster in the conditions presented to them. In this study, the fungi were asked to grow at the EC95. Surprisingly, almost no strains could grow at twice these concentrations (Fig 4a). The RFs of the fungicides known to be 10 + for DMIs, 20+ for SDHIs and 100 + for MBCs. Hence the phenotype

of the strains found in this paper are not the same as the ones typically found in the field (I concede that the area of resistance with very small RFs is understudied).

>Indeed, as we did not select field genotypes, we did not expect to find phenotypes exactly similar to field ones. Please note that the RFs you mention are usually calculated from EC50s. In Fig. 4A, we mention growth at two times the EC95, which doesn't mean that RFs would be only of 2 (depending of the shape of the dose-response curve). But indeed, mutants seem to exhibit rather low resistance, which is consistent with the non-target site resistance mechanisms which have been selected (see previous discussion on minor loci involved in resistance). Experimental evolution did not succeed here in selecting the mutations that usually arise in the field. This indicates that the field of mechanism possibilities is much wider than we imagine and does not exclude that the mutations we have identified are not selected in the field as well. If they are unknown and of minor effect, they are probably ignored by molecular monitoring even if putatively present in field populations and certainly contributing to the variability of resistance phenotypes with quantitative resistance depending on their combination. This does not mean, however, that the results of our work cannot be transposed to the field. We discussed this further.

Minor points.

The title - As I read the data, the outlier was the MBC. This is described in the paper as the high-risk fungicide. So the title could be 'Delayed but generalist resistance in alternation but only for low and medium risk fungicides.

>We do agree with this nuance of the reasoning. However, this title might sound not really appealing, as commented by a naïve native-speaker colleague in our lab, and then not prone to promote alternation as an option to manage resistance. It did not include the remark from Reviewer #2 either (see below). The title was finally revisited to ““Antifungal alternation can be beneficial for durability but at the cost of generalist resistance”, that we do think do not oversell our findings but is more positive.

The abstract and intro seek to generalise the results to all pesticides and even clinical antibiotics. I would suggest limiting the scope to agricultural fungicide.

>See previous answer to the editor. We restricted the scope to agricultural pesticides, to allow comparison with other pests.

Line 73 - the authors distinguish TSR (target site resistance) and NSTR. I would suggest that TSR is two phenomena - mutations in the coding region (TSM) and overexpression (TSO). Furthermore, there can be many TSMs in a single strain.

>Fully agree. We thought that it was clear but tried to clarify. We did not used the last acronyms, not to multiply them in the manuscript.

331 - why use 100 mg/ml strep and pen? These are high glucose low oxygen conditions.

>This is to prevent bacterial contamination, as the experiment duration is more than 3 months. This was found useful in preliminary experiments (despite the unfavourable conditions you mention), while not changing fungal growth significantly. This detail was added in M&M.

The paper is hard to read at times. The figures are packed and fonts are small. The three fungicides are C, B and P and are high, med and low risk but in Fig v3 are in panel B, C and D. I'd suggest using the acronyms CBX, PTZ and BZF in figures and the text.

>We are sorry about that. We did our best to enlarge the fonts wherever possible, but indeed, the figures are complex and it is difficult to do it without overloading them, otherwise the whole thing will become even more unreadable. It was not possible to use the acronyms because it lengthened the titles even more in all figures and obliged to reduce fonts. They were also too large to be used in the small coloured captions that mentioned the names of the strategies everywhere. But we transformed the panel letters into lower case to avoid confusion with fungicide/strategies acronyms.

119 and 163 - the sigmoid rise of resistance to PTZ and BZF. Why would this be? I am not sure this is equivalent to the emergence and selection phase.

Indeed, the term was confusing. We replaced “sigmoid dynamics” by “sigmoid curve” line 184. A sigmoid curve is a S shaped curve including typically an initial exponential phase, an approximately linear phase and finally an asymptotic phase. The observed sigmoid rise of resistance to PTZ and BZF is represented in figure 1C first graphics entitled “No alternation”.

The emergence and selection phase are defined by Hobbelen et al. 2014 as time “until the resistant strain succeeds in building up a large enough sub-population so that it is unlikely to die out due to chance” and “phase in which the application of fungicides increases the frequency of the resistant strain in the pathogen population”, respectively. The lowest frequency reliably detectable in our experiment was 15% (determined by the characteristics of our OD reader). This value was chosen to define the end of emergence phase, because at this frequency the resistant strains are unlikely to die due to chance. After having reached 90%, the frequency of resistant strain became almost constant and did no more increase. Thus, we defined the selection phase as time when frequencies were 15% and 90%.

Hobbelen, P. H., Paveley, N. D., & van den Bosch, F. (2014). The emergence of resistance to fungicides. *PLoS One*, 9(3), e91910.

168 Fig 2c is missing

>Mistake: this is Fig 2b. Thanks!

166 and 201 - the reader would be helped by a sentence or two describing the figure. Much of this text is in the figure legend but it should instead/also be in the text.

>We understand your relevant request. However, as this presentation of our results was not criticized by reviewers 2 and 3, we deliberately left the descriptions of the figures in the legends in order not to complicate the text even more and include redundancy in the manuscript. But we added sentences to explain the principle of the analysis and how to read the figures (l189-l191; l226-228).

Reviewer #2:

This paper uses experimental evolution to test fungicide resistance evolution under different management strategies; specifically, the alternation of different chemical modes of action. The authors clearly explain that their findings will be relevant to any cases of resistance evolution and are therefore of wider interest beyond fungicides and plant pathology.

In all cases, resistance was selected more slowly under alternating fungicides of different modes of action than with repeated use of a single compound. However, the rate of resistance evolution per treatment with a given compound was only reduced in alternations with fungicides of lower resistance risk.

>Yes indeed, but it was not increased in alternations including the high risk-fungicide. It was similar to that to that of the same fungicide when used alone (Fig. S1). Therefore, alternation was never “a bad choice” to use alternation, whatever the fungicides. This is the positive message that we tried to highlight more in the revised version of this manuscript.

They also observe greater cross-resistance between different fungicides in lines selected with alternating fungicides, compared to more specialised resistance selected with a single mode of action; interestingly this extends to additional modes of action not used within the alternation. This paper presents some interesting data, although in some cases the interpretation is somewhat speculative and not all alternative explanations have been considered. In particular, where the effectiveness depends on the resistance risk of the alternation partner, and where selection varies over time as the populations evolve, this will be partly because as resistance evolves against the alternation partner, treatments with the alternation partner fungicide will result in smaller-to-no reductions in microbial population size; and in this experimental system, this will alter the proportion of selected versus ‘immigrating’ population going into the next round of selection with the focal fungicide (with immigration diluting the effects of previous rounds of selection). The effects of basic population dynamics need to be accounted for before invoking hypotheses about adaptive landscapes.

>Indeed, this is an assumption that we missed to discuss. These comments have been added in the discussion (I299). Thanks!

However, as mentioned in I323 and in Fig. 3, the discussion about fitness landscapes concerns lines having undergone the continuous use of the same fungicide. The previous comment about population dynamics may not be appropriate as it relies on the relative efficacy of at least two fungicide partners. We have not found any other assumption than fitness landscapes to explain the changes in population composition observed in sequence regimes – the dilution from control lines may decrease the selection rate of resistance but not influence the phenotypic diversity in population, as observed here. But we would be pleased to discuss any other idea.

It should also be noted that, with the exception of Beta tubulin E198K, the evolving lines did not evolve any of the specific resistance mechanisms that have been reported in field populations. Whilst this is not a fatal flaw in a paper using *Z. tritici* as a model for general principles, how the various findings may or may not relate to field conditions should be discussed. For example, mixtures/alternations are widely used and yet target site specific resistance is common.

> Indeed, this is a major criticism addressed by several referees. Our objective in this paper was to dissect the components of selection for the various strategies in a general manner, with no *a priori* on the resistance mechanisms that might be selected, as some unknown mutations might be selected under field conditions as well. In our mind, this work does not aim at predicting which mutations will arise in the field under the selection from contrasting regimes. But this work is rather a simplified model to understand how resistance is selected and eventually predict how we can play with selection components to slow down resistance, from an operational point of view. This is why we had not investigated further the side topic of resistance mechanisms. Nevertheless, we acknowledge that deciphering resistance mechanisms in our experiment is relevant for the comparison with resistance evolution in field conditions and might be also of academic interest, as it may highlight original unsuspected mechanisms. Therefore, we carried out additional experiments to search for length polymorphisms in the promoters of the three target genes of the fungicides used in this experiment and known to carry mutations in the field. We also measured the expression of the same genes and of *mfs1* (involved in the MDR phenotype) in a selection of isolates. This conducted us to design and validate some protocols that were not available and we do apologize for the subsequent delay when submitting a revised version of this manuscript. Altogether, this information is provided in the new Methods S2 and S3 and Figure S2. We found that enhanced efflux due to the overexpression of *mfs1* might indeed be at work in some of the isolates (see additional comments in M&M and results sections) but that it did not explain the resistance phenotype of all selected isolates. The unknown changes that we have selected may also be selected in the field, in addition to TSR, from mixture/alternation applications; they just haven't been looked for and this discrepancy doesn't imply that our findings cannot be transposed to field conditions. But we have discussed this further. Further investigation, including genome sequencing, would be needed to elucidate these pending resistance mechanisms. But this would deserve a whole specific paper, with objectives too distant from our own. This is why we mentioned and discussed these further investigations only as perspectives.

The reason why mixtures/alternations did not prevent the selection of (target site) resistance in the field might be because we do not use them in an optimal manner. Indeed, for historical reasons, MoAs were used successively, which means that at least one fungicide was not used on fully susceptible populations, i.e. at its optimal efficacy. This explains the emergence of multiple resistance (the new resistance is selected in backgrounds already resistant to the previous fungicide). In another work to be submitted soon, we demonstrated that both mixture and alternation were not sustainable in the long term when multiple resistance was present in the population, even at low frequency, whereas both strategies were effective when only single resistances were present).

Specific comments

Abstract:

line 28-29: The relative intrinsic resistance risk determines the relative benefit of alternation for each fungicide; I'm not sure they demonstrate a relationship between the relative resistance risk of components and the level of generalism in the evolved lines?

>Indeed, the relative risk of resistance of a fungicide may certainly determine the rate of resistance evolution (the "quantitative performance" of the strategy), and the choice of the AI, i.e. its nature, may determine the resistance mechanisms (the "qualitative performance" of the strategy). The relative risk of resistance is intrinsically linked to the AI, this is why we had summarized this conclusion in this way. We added this detail.

Introduction:

The introduction gives a broad overview of wider questions but would benefit in places from a little more detail and some specific examples.

>This was addressed while focusing on agricultural pesticides, as also recommended by the other referees, as we found useful to keep examples and references on other pests.

Line 58: What are some key areas of disagreement between models and empirical data? Which biological traits could affect the results?

>The ranking of the efficacy of the strategies differs among approaches. This was precised (l61), as well as the reasons for these discrepancies (l62-64).

67: phenotypic impact?

>Not sure of the problem you mention. We added that the characteristics of the resistant individuals selected are more rarely made explicit.

73: levels of cross resistance for TSR vary between mutations/ modes of action

>Added

83: "any given strategy": examples?

>Alternation, mixture or mosaic. Detailed.

96: explain what strategies and how they impacted loss of efficacy?

>Detailed

97: the cited references cover several European countries, not just France

>The citations from foreign countries also cover the beginning of the sentence, i.e. the evolution of resistance to multiple MoAs. We grouped everything at the end not to interrupt the sentence.

99: explain "not associated to TSR": TSR and MDR can both be found in the same isolates

>Added

104: define "alternation rhythm" at the first use of the term

>Added in the first paragraph of the results section.

105: what do you mean by "interplay"? patterns of cross resistance? relative resistance risk of each?

Relative resistance risk of the alternation partner affected the rate of resistance evolution but in what ways did it alter the nature of the trade-off with generalism?

>There might be a misunderstanding here, as the components (of alternation) meant the drivers (of this strategy), and not the alternated AIs. We changed this sentence.

106: qualitative and quantitative: does this mean speed versus generalism of resistance?

> The other way around. This is detailed in the results but we added this information at the end of the introduction).

Results:

The authors should further clarify, ideally within the section headings, the distinction between total time to resistance (for which alternations were always neutral to beneficial) and rate of selection during treatment with a given fungicide (for which only alternation with lower resistance risk fungicides was beneficial), otherwise the stated findings appear contradictory. See also abstract line

27: “slows the evolution of resistance” could be misleading.

>We realized that while trying to avoid repeats in the redaction, we may have used different wordings for the same thing. In the first sub-section of the results (Fig 1), we discuss about the “overall resistance evolution rate ρ ”, calculated over all cycles. This growth is a speed, expressed as number of cells/cycle, and not a duration, as would be suggested while using the words “total time to resistance”, as suggested.

In the second paragraph of the Results and figure 2, we introduced the same index but calculated only over the cycles with exposure of the fungicide pf. To avoid the confusion, we introduced the wording “fungicide-specific resistance evolution rate” when pf was designated. Changes were made thoroughly wherever needed, including subheadings.

121: define “generalised” at the first use of the term (especially given potential confusion between generalisation and generalism)

>Generalisation is the time (in cycles) needed for the mean Malthusian growth to reach at least 90% of that of the control line. This definition was added but we replaced the words generalized/generalization by “established/establishment” throughout the manuscript as recommended by reviewer 3.

122: in which conditions did resistance never generalise?

>When low- to medium risk fungicides were used. Added.

130: decrease in selection compared to the lower or higher risk fungicide alone?

>Yes. Detail added.

140: “multi-directional selection”: in this case do you mean selection per treatment with the fungicide in question?

>We mean the “diversification of selection pressure” by several AIs in alternation regimes and used this term not to repeat the previous sentence.

151: this is a key point and should be further highlighted: do alternations have a net benefit on longevity of the total set of fungicides, or is it simply a zero-sum combination of resistance selection for the different modes of action?

>Indeed. This point is highlighted in the abstract and largely mentioned in the discussion (l282 and onwards).

162 and following section: Are these changes over time all attributable to the adaptive landscape of the fungicide in question, or does the effect of alternation change as resistance also evolves against the alternating partner? ie. in the early generations, the alternating partner is highly effective at reducing pathogen population size, necessitating a larger input via immigration from the non-selected line. In the later generations, especially for a higher-risk mixing partner, treatment with that fungicide has little effect and is more like a treatment gap in which the population is left to grow.

>Please see our previous answer on this topic. As mentioned previously, this section describes the evolution of resistance patterns in regimes under continuous exposure. We restricted this analysis only to sequence regimes to avoid confounding effects that may have occurred when examining evolutionary trajectories from alternation regimes. Indeed, it would have been nearly impossible to dissect how which fungicide would select for which phenotype after such complex selection. Avoiding this confounding effect would have required to isolate and phenotype thousands of strains from each cycle to have a clear view of adaptive landscapes, which was not technically feasible. Samples might be analysed further once molecular tools are made available. To avoid any misunderstanding and make a transition with the next subsection, we added a final sentence about the expectations in selection regimes.

182: do isolates from different time points show sufficiently distinct resistance profiles to suggest that different genotypes were selected over time? If so, please add figures demonstrating this.

>This is shown in Fig 3. Please note that droplet tests were achieved on single isolates, and not on populations. Changes in scores then reflect the variation of phenotypes (and putatively on genotypes) and are not biased by changes in resistance frequency, as it might have been expected while testing populations. Over time, some resistance (e.g. to tolnaftate, boscalid or azoxystrobin) were selected and sometimes counterselected. As these fungicides were not used for the selection, we might conclude that these phenotypes are associated to specific genotypes. Changes in the frequency of resistance profiles was found significant over time (see legend of Fig. 3).

220: can beta-tubulin genotypes be tracked over time (re. previous point)?

>Indeed, this could be possible pending extensive sequencing of isolated strains. Nevertheless, we found the tub2 E198K change only occasionally, and the number of isolated strains (e.g. 12 per selection regimes) is limited. Over-time isolations were performed only for sequence regimes. For all

these reasons, the estimation of the frequencies of beta-tubulin changes might not be really informative being given our sampling. NGS on population might be useful here to describe fitness landscapes properly, as it would include not only target-site changes but whole-genome changes. But this would be an independent (planned) project.

235: why specifically “local” populations?

>In the field, selection might differ locally, as fungicide programs (i.e. selection regimes, differing for their drivers) may change from field to field.

237: again, is there a net benefit of heterogeneity here, or only zero-sum sharing of resistance selection across alternation components?

>This is long discussed a few lines later.

246: clarify: total time to resistance, not selection rate during treatment

>See previous comment. The two statements refer to indexes differing in their nature.

268: There should also be some discussion of alternations versus mixtures

>We added a general sentence but do not want to expand this discussion further without being speculative: this is difficult to compare the two strategies, as we did not test mixture in this paper. A fair comparison is being achieved in another manuscript close to submission.

280-283: clarify whether you are referring to fungicides with negative resistance or just with no cross-resistance. I think you mean the trade-off between high resistance to a single MOA versus lower-level resistance to multiple MOAs. The point here is not necessarily that the fitness landscape under selection by any one fungicide is more rugged, but that the landscape shifts between one fungicide and another, favouring genotypes with a higher average fitness over time.

>Indeed, we mean fungicide with no cross-resistance. Your point was added.

285: What is the evidence for additivity or synergy? The wider cross-resistance is an example of pleiotropy not epistasis.

>Agree. This was deleted.

296-300: Is this due to fitness costs or resistance levels of resistance mutations against the fungicide in question, or is it confounded by the relative effectiveness of the alternation partner over time? A higher risk fungicide will have more effect during the early stages, before resistance to the higher-risk fungicide evolves and it stops effectively reducing the pathogen populations; whereas a lower-risk mixing partner will continue to delay resistance even once resistance to the high-risk fungicide is in the final stages of selection.

>This may definitely reflect the balance between the relative fitness costs and levels of resistance of selected mutations for each AI, as we know that they vary accordingly in each single species. They are

the primary determinants of the evolution of resistance. But indeed, there might be a confounding effect with the relative effectiveness of the partners over time, as it directly reflects the population composition, i.e. the relative frequency of each genotype, which changes over time, due to the selection, and also to immigration from control lines, in some situations. In our experimental design, the initial size of the population was constant (i.e. 10^7 cells) in order to prevent line extinction due to very tight bottlenecks, as promoted by high relative effectiveness of both compounds in the first cycles of the experiment. As you previously noticed, before resistance to the alternation partner evolved, the size of the population at the end of the cycle was very low and we had to add some immigrating cells, which reduced the initial resistance frequency in some situations. This favoured equally the relative effectiveness of both AIs, as the dilution was achieved with susceptible cells. In the situations where immigration was not needed, no susceptible cells were introduced so the effectiveness of both AIs were equally “disadvantaged” by the lack of dilution. Then, even though immigration slowed down the increase of resistance frequency to each partner in the first cycles of the experimental evolution, there was no bias between AIs, as their resistance dynamics was modified similarly (i.e. same “dilution rate”). But the relative effectiveness of each partner definitely determined the variation in resistance frequency over the cycles that you describe above. Practically, we did not observe exactly these variations in Fig2b for all pairs of fungicides. This more complex situation might be because multiple genotypes, with different intrinsic features, are co-selected in alternation regimes, which is also supported by the droplet analysis. Although we observed the fungicide-specific growth rate at five different time-courses regularly covering the whole range of resistance frequency, we may also have missed critical threshold values relevant for a particular interaction.

308: When discussing the relevance to growers, again there needs to be discussion of using multiple MOAs in alternations versus mixtures.

>As mentioned previously, we would prefer to keep this discussion for our on-going paper studying the direct comparison between the two strategies.

322: Can relative risk be adjusted by using different doses or frequencies of each partner, or using some alternation partners in mixtures?

>Yes, indeed. This sentence was added.

Methods

376: the predicted risk levels for each fungicide group are also supported by resistance evolution in the field

>The predicted risk levels mentioned here are those provided by FRAC (ref 74). These proposals are generally made prior to the use of new AI/MoA.

Indeed, resistance evolution is the best proxy for the risk level but it is an *a posteriori* assessment. Some discrepancies were noticed between the FRAC prediction and empirical situations (Grimmer et al, 2014). For this reason, we preferred to keep the “officially predicted” risk assessments (as mentioned in the reference) but we think that the *a posteriori* assessed risk is long discussed in this manuscript.

393-394: Are “control lines” and “control solvent lines” the same or were there control lines with and without solvent?

>These were two different controls. Control lines (without solvent and fungicides) were maintained as a backup for immigration, in case the control solvent was contaminated (which never happened) and to assess the impact of 0.5% ethanol on growth. We found no significant difference for the growth of these two controls.

409: was the rate of genetic drift tested?

>This was not possible because lines generally extinguished, without immigration, before resistance was suspected (by OD change).

476: Need to discuss the assumption that different phenotype scores correspond to different alleles: could there be polygenic resistance? Multiple alleles with similar phenotypes?

>Indeed, differing scores for a given testing modality could be also different combinations of the same alleles. This was added. However, as we used a high number of testing modalities, and especially fungicides not used for the selection, we hypothesized that multiple alleles might be preferentially selected in distinct resistant profiles.

Figures

1b: Is the triangle illustration needed? It may be clearer just to use a simple legend for the 3 individual AIs

>The triangle is used here to show the pairs/trio of alternated AIs, and also to show the colors/labels associated to these combinations. We think that losing this information might be detrimental for the general understanding, but we tried to be clearer in the legend. Letters mentioning the sub-panels were changed to lower-case, to avoid confusion with fungicides names.

Reviewer #3:

General comment

Short: An excellent piece of work put together. I do not have many comments.

My main critic is that the introduction part – in contrast to the rest of the manuscript – is hard to read and needs to be rephrased before the manuscript should be accepted for publication. Title: Consider rephrasing the title to give a clear idea on your findings, e.g., ‘Alternation of antifungal modes of action confer general fungicide resistance.

The introduction is hard to read, especially the first part. Please rephrase the first paragraph with shorter sentences and be more specific when using terms such as resistance (fungicide resistance), strategies (spray strategies; L 50 “Strategies to delay fungicide resistance...”). I see the point in starting from the more general point of pesticide/drug resistance and the theories around different usage pattern selecting differently. However, very much information is described very

condensely and makes is hard understand. I would suggest sticking to pesticides used in agriculture, zooming in on the *Z. tritici* model, and finishing off that these results can be transferred to human and animal health.

Give a short introduction why *Z. tritici* is a good model to assess fungicide resistance evolution.

>The introduction and title were rephrased. In particular, we justified the choice of *Z. tritici* (and again in the discussion) as a model to address our question.

L 38: Human societies = humanity?

>Done

L 82: "..., for which increasing use of different MoA/herbicides"

>Done

L 96: "...mutations and enhanced efflux; the latter being a generalist mechanism causing..."

>Sentence shortened.

L 98: yeast.like form = blastospores?

>Added

Drop the term "agents", and introduce AI sooner. Introduce and use the abbreviation MoA already in the introduction.

>Done

Results:

L 115: 'overall' instead of 'global'

>Done

L 121: "... 1.4 times higher with C than with P; these significant differences reflect ..."

>Done

L 123: " The overall resistance rate..."

>Done

L 162: Isn't 90% a high value. I would say that resistance already is generalized with 75%

>Indeed, the appreciation of the threshold for generalisation may differ according to experts and resistance cases, as loss of efficacy might be observed starting at different levels. We called it a "established-resistance" but kept this value, as we would have had to perform models and figures again for another value. Please see also other comments on this topic.

L 242: "...in terms of delaying overall fungicide resistance...."

>Changed

Methods

L 325: Headline to general: what ancestral population, what kind of assessment?

>Added

L 328: Is taking the isolate out of – 80C to let it thaw at room temp gentle?

>Yes! Changed

L 336: delete the second "RH"

>Done

L 350: Why was carbendazim used?

>We needed a high-risk fungicide (to explore the range of resistance risks), ideally with knowledge available for field resistance. We had the choice between benzimidazoles and QoIs. We excluded QoIs because target-site resistance was hardly selected *in vitro* in literature for QoIs, whereas we found it was possible for benzimidazoles. This choice was found successful in this experiment, with the isolation of the E198K change in beta-tubulin.

L 354: associated with...

>Done

L 358: OD 405

>Done

L 382: 4×10^5 sp. mL⁻¹

>Done

L 386: "...were not treated with any fungicide"

>Done

L 407: Reference for the Malthusian model?

We added the reference: Fisher, R. A. The genetical theory of natural selection: A complete variorum edition (University Press, 1999).

L 429: Isn't 75% already a high % for intermediate resistance frequency?

>Indeed (see previous comment). Thresholds are critical conventions and were defined according to the technical limitations of our OD reader, to guarantee repeatable and reliable results (I184). But the most important is to explore the wide range of frequencies, whatever the limits between the phases.

L 430: Personally, I am not a fan of the word "generalization" in this context. How about 'established resistance' or 'full resistance'.

>Changed to established resistance.

L 440: "...10⁵, and ..."

>Done

L 453-454: You had four droplets on each AI – if growth was observed, it was scored 1. If all droplets grew, it got the score 4. Is that correct?

>Yes, indeed. Scores ranged between 0 and 4, for each modality.

L 492-494: Why not from all isolates?

>This was our first idea, but it was too expensive. We gave it up without regret when we noticed that target site mutations were very rare or absent in most lines, after the sequencing of the first batches.

Figures:

The figures are very illustrative and helpful to understand the experimental design.

>Thanks!

Ref.:

L 695: Is this the right way to cite a book

>Reference completed

Reviewers' comments:

Reviewer #2 (Remarks to the Author):

The authors have made most of the changes suggested in previous reviews, resulting in a greatly improved manuscript.

However, from my previous comments: "In particular, where the effectiveness depends on the resistance risk of the alternation partner, and where selection varies over time as the populations evolve, this will be partly because as resistance evolves against the alternation partner, treatments with the alternation partner fungicide will result in smaller-to-no reductions in microbial population size; and in this experimental system, this will alter the proportion of selected versus 'immigrating' population going into the next round of selection with the focal fungicide (with immigration diluting the effects of previous rounds of selection). The effects of basic population dynamics need to be accounted for before invoking hypotheses about adaptive landscapes":

The authors have added a sentence to this effect in the discussion but not amended their claims in the results section.

Overall, the selection dynamics over time should first and foremost be interpreted in the context of:

- 1) Initial stages: no resistance to either fungicide
- 2) Intermediate points: resistance to focal fungicide but not mixing partner (if mixing partner is lower risk) OR resistance to mixing partner but not focal fungicide (if mixing partner is higher risk) OR possibly partial resistance against both (if risk levels are similar)
- 3) Late stages / final time point: resistance to both fungicides.

Any effects of the lineage exploring different parts of the adaptive landscape over time would need to be apparent over and above (or deviating from) the expected shifts in selection from the above scenarios. Perhaps the shift from the second to the third resistance profile from the top in figure 3c shows this, with growth on B alone going down whilst growth on B+C goes up, but it is not clear from phenotypic data whether this reflects the overall pattern. Therefore, I would prefer the subheading in line 181-182 to make a less specific claim, e.g. "Resistance selection differs over the course of evolution, with shifts in the interplay between mixture partners", with adaptive landscape exploration then presented as one possible factor involved.

With this minor further revision the paper will, in my opinion, be suitable for publication.

Answer to reviewers

Reviewer #2 (Remarks to the Author):

The authors have made most of the changes suggested in previous reviews, resulting in a greatly improved manuscript.

However, from my previous comments: "In particular, where the effectiveness depends on the resistance risk of the alternation partner, and where selection varies over time as the populations evolve, this will be partly because as resistance evolves against the alternation partner, treatments with the alternation partner fungicide will result in smaller-to-no reductions in microbial population size; and in this experimental system, this will alter the proportion of selected versus 'immigrating' population going into the next round of selection with the focal fungicide (with immigration diluting the effects of previous rounds of selection). The effects of basic population dynamics need to be accounted for before invoking hypotheses about adaptive landscapes":

The authors have added a sentence to this effect in the discussion but not amended their claims in the results section.

Overall, the selection dynamics over time should first and foremost be interpreted in the context of:

- 1) Initial stages: no resistance to either fungicide
- 2) Intermediate points: resistance to focal fungicide but not mixing partner (if mixing partner is lower risk) OR resistance to mixing partner but not focal fungicide (if mixing partner is higher risk) OR possibly partial resistance against both (if risk levels are similar)
- 3) Late stages / final time point: resistance to both fungicides.

>This explanation about the impact of population dynamics on the selection rate was added in the results section (193-202).

Any effects of the lineage exploring different parts of the adaptive landscape over time would need to be apparent over and above (or deviating from) the expected shifts in selection from the above scenarios. Perhaps the shift from the second to the third resistance profile from the top in figure 3c shows this, with growth on B alone going down whilst growth on B+C goes up, but it is not clear from phenotypic data whether this reflects the overall pattern. Therefore, I would prefer the subheading in line 181-182 to make a less specific claim, e.g. "Resistance selection differs over the course of evolution, with shifts in the interplay between mixture partners", with adaptive landscape exploration then presented as one possible factor involved.

>This subtitle was changed to 'Resistance selection differs over the course of evolution, reflecting contrasting shifts in the interplay between fungicides and phenotype-adaptive landscapes' (171-172).

With this minor further revision the paper will, in my opinion, be suitable for publication.

>In addition to the changes mentioned before, we took benefit of this new round of revision to

edited minor typo or rewrite some sentences more clearly, without changing their content (visible with track change).